# Learning to Compose Visual Relations

**Nan Liu** [*]
University of Michigan
liunan@umich.edu

**Shuang Li** [*]
MIT CSAIL
lishuang@mit.edu

**Yilun Du** [*]
MIT CSAIL
yilundu@mit.edu

**Joshua B. Tenenbaum**
MIT CSAIL, BCS, CBMM
jbt@mit.edu

**Antonio Torralba**
MIT CSAIL
torralba@mit.edu

## Abstract

The visual world around us can be described as a structured set of objects and their associated relations. An image of a room may be conjured given only the description of the underlying objects and their associated relations. While there has been significant work on designing deep neural networks which may compose individual objects together, less work has been done on composing the individual relations between objects. A principal difficulty is that while the placement of objects is mutually independent, their relations are entangled and dependent on each other. To circumvent this issue, existing works primarily compose relations by utilizing a holistic encoder, in the form of text or graphs. In this work, we instead propose to represent each relation as an unnormalized density (an energy-based model), enabling us to compose separate relations in a factorized manner. We show that such a factorized decomposition allows the model to both generate and edit scenes that have multiple sets of relations more faithfully. We further show that decomposition enables our model to effectively understand the underlying relational scene structure. Project page at: https://composevisualrelations.github.io/

## 1 Introduction

The ability to reason about the component objects and their relations in a scene is key for a wide variety of robotics and AI tasks, such as multistep manipulation planning [11], concept learning [25], navigation [43], and dynamics prediction [3]. While a large body of work has explored inferring and understanding the underlying objects in a scene, robustly understanding the component relations in a scene remains a challenging task. In this work, we explore how to robustly understand relational scene description (Figure 1).

Naively, one approach towards understanding relational scene descriptions is to utilize existing multi-modal language and vision models. Such an approach has recently achieved great success in DALL-E [36] and CLIP [35], both of which show compelling results on encoding object properties with language. However, when these approaches are instead utilized to encode relations between objects, their performance rapidly deteriorates, as shown in [36] and which we further illustrate in Figure 7. We posit that the lack of *compositionality* in the language encoder prevents it from capturing all the underlying relations in an image.

To remedy this issue, we propose instead to *factorize* the scene description with respect to each individual relation. Separate models are utilized to encode each individual relation, which are then subsequently *composed* together to represent a relational scene description. The most straightforward

---

[*]indicates equal contribution

35th Conference on Neural Information Processing Systems (NeurIPS 2021).

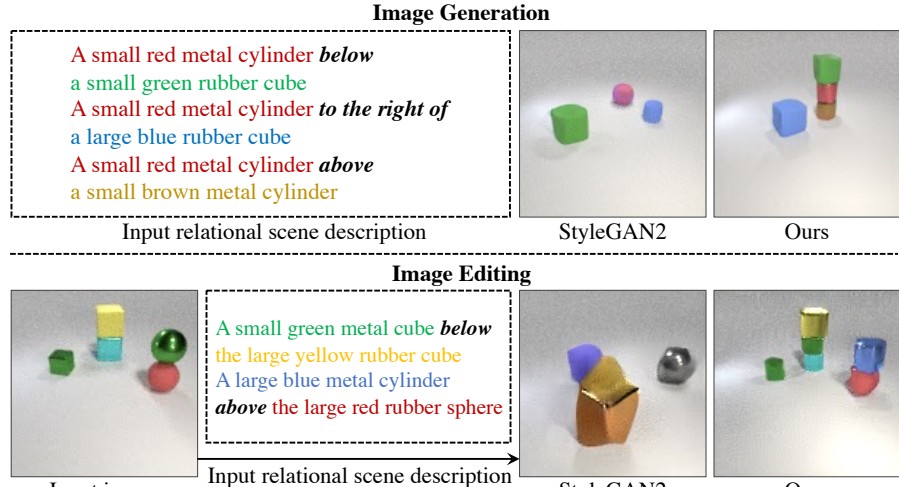

Figure 1: Our model can generate and edit images with multiple composed relations. **Top**: Image generation results based on relational scene descriptions. **Bottom**: Image editing results based on relational scene descriptions.

approach is to specify distinct regions of an image in which each relation can be located, as well as a composed relation description corresponding to the combination of all these regions.

Such an approach has significant drawbacks. In practice, the location of one pair of objects in a relation description may be heavily influenced by the location of objects specified by another relation description. Specifying a priori the exact location of a relation will thus severely hamper the number of possible scenes that can be realized with a given set of relations. Is it possible to factorize relational descriptions of a scene and generate images that incorporate each given relation description simultaneously?

In this work, we propose to represent and factorize individual relations as unnormalized densities using Energy-Based Models. A relational scene description is represented as the product of the individual probability distributions across relations, with each individual relation specifying a separate probability distribution over images. Such a composition enables interactions between multiple relations to be modeled.

We show that this resultant framework enables us to reliably capture and generate images with multiple composed relational descriptions. It further enables us to edit images to have a desired set of relations. Finally, by measuring the relative densities assigned to different relational descriptions, we are able to infer the objects and their relations in a scene for downstream tasks, such as image-to-text retrieval and classification.

There are three main contributions of our work. First, we present a framework to factorize and compose separate object relations. We show that the proposed framework is able to generate and edit images with multiple composed relations and significantly outperforms baseline approaches. Secondly, we show that our approach is able to infer the underlying relational scene descriptions and is robust enough in understanding semantically equivalent relational scene descriptions. Finally, we show that our approach can generalize to a previously unseen relation description, even if the underlying objects and descriptions are from a separate dataset not seen during training. We believe that such generalization is crucial for a general artificial intelligence system to adapt to the infinite number of variations of the world around it.

## 2 Related Work

**Language Guided Scene Generation.** A large body of work has explored scene generation utilizing text descriptions [13, 17, 20, 30, 34, 36, 38–40, 46–48]. In contrast to our work, prior work [17, 38, 47, 48] have focused on generating images given only a limited number of relation descriptions. Recently, [36] shows compelling results on utilizing text to generate images, but also explicitly state that relational generation was a weakness of the model. In this work, we seek to tackle how we may generate images given an underlying relational description.

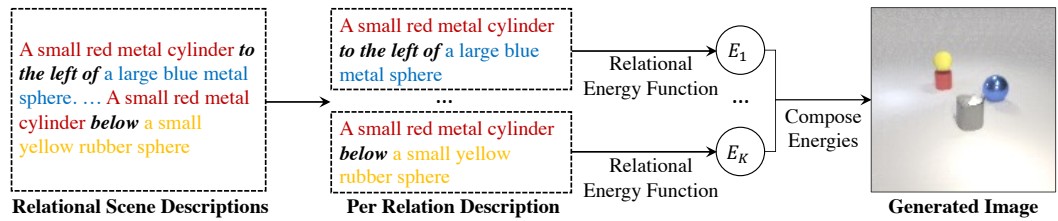

Figure 2: Overview of our pipeline for understanding a relational scene description. A relational scene description is split into a set of underlying relation descriptions. Individual relation descriptions are represented as EBMs which are subsequently composed together to generate an image.

**Visual Relation Understanding.** To understand visual relations in a scene, many works applied neural networks to graphs [1–3, 5, 14, 15, 18, 20, 28, 29, 32, 37]. Raposo et al. [37] proposed Relation Network (RN) to explicitly compute relations from static input scenes while we implicitly encode relational information to generate and edit images based on given relations and objects. Johnson et al. [20] proposed to condition image generation on relations by utilizing scene graphs which were further explored in [1, 14, 15, 18, 29, 31, 44]. However, these approaches require excessive supervisions, such as bounding boxes, segmentation masks, etc., to infer and generate the underlying relations in an image. Such a setting restricts the possible combinations of individual relations in a scene. In our work, we present a method that enables us to generate images given only a relational scene description.

**Energy Based Models.** Our work is related to existing work on energy-based models [6, 8, 10, 12, 22, 33, 42, 45]. Most similar to our work is that of [7], which proposes a framework of utilizing EBMs to compose several object descriptions together. In contrast, in this work, we study the problem of how we may compose relational descriptions together, an important and challenging task for existing text description understanding systems.

## 3 Method

Given a training dataset $\mathcal{C} = \{\boldsymbol{x}_i, R_i\}_{i=1}^N$ with $N$ distinct images $\boldsymbol{x}_i \in \mathbb{R}^D$ and associated relational descriptions $R_i$, we aim at learning the underlying probability distribution $p_\theta(\boldsymbol{x}|R)$ — the probability distributions of an image given a corresponding relational description. To represent $p_\theta(\boldsymbol{x}|R)$, we split a relational description $R$ into $K$ separate relations $\{r_1 \cdots, r_K\}$ and model each component relation separately using a probability distribution $p_\theta(\boldsymbol{x}|r_k)$ which is represented as an Energy-Based Model. Our overall scene probability distribution is then modeled by a composition of individual probability distributions for each relation description $p_\theta(\boldsymbol{x}|R) \propto \prod_k p_\theta(\boldsymbol{x}|r_k)$.

In this section, we give an overview of our approach towards factorizing and representing a relational scene description. We first provide a background overview of Energy-Based Models (EBMs) in Section 3.1. We then describe how we may parameterize individual relational probability distributions with EBMs in Section 3.2. We further describe how we compose relational probability distributions to model a relational scene description in Section 3.3. Finally, we illustrate downstream applications of our relational scene understanding model in Section 3.4.

### 3.1 Energy-Based Models

We model each relational probability distribution utilizing an Energy-Based Model (EBM) [6, 26]. EBMs are a class of unnormalized probability models, which parameterize a probability distribution $p_\theta(\boldsymbol{x})$ utilizing a learned energy function $E_\theta(\mathbf{x})$:

$$p_\theta(\mathbf{x}) \propto e^{-E_\theta(\mathbf{x})}. \tag{1}$$

EBMs are typically trained utilizing contrastive divergence [16], where energies of training data-points are decreased and energies of sampled data points from $p_\theta(\boldsymbol{x})$ are increased. We adopt the training code and models from [8] to train our EBMs. To generate samples from an EBM $p_\theta(\boldsymbol{x})$, we utilize MCMC sampling on the underlying distribution, and Langevin dynamics, which refines a data sample iteratively from a random noise:

$$\tilde{\mathbf{x}}^m = \tilde{\mathbf{x}}^{m-1} - \frac{\lambda}{2}\nabla_{\mathbf{x}}E_\theta(\tilde{\mathbf{x}}^{m-1}) + \omega^m, \ \omega^m \sim \mathcal{N}(0, \sigma) \tag{2}$$

where $m$ refers to the iteration and $\lambda$ is the step size, utilizing the gradient of the energy function with respect to the input sample $\nabla_{\mathbf{x}} E_\theta$, and $\omega^m$ is sampled from a Gaussian noise. EBMs enable us to naturally compose separate probability distributions together [7]. In particular, given a set of independent marginal distributions $\{p_\theta^i(\boldsymbol{x})\}$, the joint probability distribution is represented as:

$$\prod_i p_\theta^i(\boldsymbol{x}) \propto e^{-\sum_i E_\theta^i(\boldsymbol{x})}, \tag{3}$$

where we utilize Langevin dynamics to sample from the resultant joint probability distribution.

## 3.2 Learning Relational Energy Functions

Given a scene relation $r_i$, described using a set of words $\{w_i^1, \ldots w_i^n\}$, we seek to learn a conditional EBM to model the underlying probability distribution $p_\theta(\boldsymbol{x}|r_i)$:

$$p_\theta(\boldsymbol{x}; r_i) \propto e^{-E_\theta^i(\boldsymbol{x}|\text{Enc}(r_i))}, \tag{4}$$

where $p_\theta(\boldsymbol{x}|r_i)$ represents the probability distribution over images given relation $r_i$ and $\text{Enc}(r_i)$ denotes a text encoder for relation $r_i$.

The most straightforward manner to encode relational scene descriptions is to encode the entire sentence using an existing text encoder, such as CLIP [35]. However, we find that such an approach cannot capture scene relations as shown in Supplement Section A. An issue with such an approach is that the sentence encoder loses or masks the information captured by the relation tokens in $r_i$.

To enforce that the underlying relation tokens in $r_i$ is effectively encoded, we instead propose to decompose the relation $r_i$ into a relation triplet $(r_i', o_i^1, o_i^2)$, where $r_i'$ is the relation token, *e.g.* "below", "to the right of", $o_i^1$ is the description of the first object, and $o_i^2$ is the description of the second object appeared in $r_i$. Each separate entry in the relation triplet is then separately embedded.

Such an encoding scheme encourages the models to encode underlying objects and relations in a scene, enabling us to effectively model the relational distribution. We explored two separate approaches to embed the underlying object descriptions into our relational energy function.

**CLIP Embedding.** One approach we consider is to directly utilize CLIP to obtain the embedding of an object description. Such an approach may potentially enable us to generalize relations in a zero-shot manner to new objects by utilizing CLIP's underlying embedding of the object, but may also hurt learning if the underlying object embedding does not distinctly separate two object descriptions.

**Random Initialization.** Alternatively, we may encode an object description using a learned embedding layer that is trained from scratch. In this approach, we extract a scene embedding by concatenating the learned embeddings of color, shape, material, and size of any two objects, $o_i^1, o_i^2$, and their relation $r_i$.

## 3.3 Representing Relational Scene Descriptions

Given an underlying scene description $R$, we represent the underlying probability distribution $p(\boldsymbol{x}|R)$ by factorizing it as a product of probabilities over the the underlying relations $r_k$ inside $R$.

Given the separate relational energy functions learned in Section 3.2, this probability $p(\boldsymbol{x}|R)$ is proportional to

$$p(\boldsymbol{x}|R) = e^{-E_\theta^k(\boldsymbol{x}|R)} \propto \prod_k p(\boldsymbol{x}|r_k) = e^{-\sum_k E_\theta^k(\boldsymbol{x}|\text{Enc}(r_k))}, \tag{5}$$

which is a new EBM with underlying energy function $E_\theta(\boldsymbol{x}|R) = \sum_k E_\theta^k(\boldsymbol{x}|\text{Enc}(r_k))$. The overview of the proposed method is shown in Figure 2, where $E_1, \cdots, E_K$ correspond to the $K$ individual relational energy functions $E_\theta^k(\boldsymbol{x}|\text{Enc}(r_k))$.

## 3.4 Downstream Applications

By learning the probability distribution $p(\boldsymbol{x}|R)$ with corresponding EBM $E_\theta(\boldsymbol{x}|R)$, our model can be applied to solve many downstream applications, such as image generation, editing, and classification, which we detail below and validate in the experiment section.

Table 1: Evaluation of the accuracy of object relations in the generated images or edited images on the CLEVR and iGibson datasets. We compare our method with baselines on three test sets, *i.e. 1R*, *2R*, and *3R* (see text).

| Dataset | Model | Image Generation (%) | | | Image Editing (%) | | |
|---|---|---|---|---|---|---|---|
| | | 1R Acc | 2R Acc | 3R Acc | 1R Acc | 2R Acc | 3R Acc |
| CLEVR | StyleGAN2 | 10.68 | 2.46 | 0.54 | 10.04 | 2.10 | 0.46 |
| | StyleGAN2 (CLIP) | 65.98 | 9.56 | 1.78 | - | - | - |
| | Ours (CLIP) | 94.79 | 48.42 | 18.00 | 95.56 | 52.78 | 16.32 |
| | Ours (Learned Embed) | **97.79** | **69.55** | **37.60** | **97.52** | **65.88** | **32.38** |
| iGibson | StyleGAN2 | 12.46 | 2.24 | 0.60 | 11.04 | 2.18 | 0.84 |
| | StyleGAN2 (CLIP) | 49.20 | 17.06 | 5.10 | - | - | - |
| | Ours (CLIP) | 74.02 | 43.03 | **19.59** | 78.12 | 32.84 | 12.66 |
| | Ours (Learned Embed) | **78.27** | **45.03** | 19.39 | **84.16** | **44.10** | **20.76** |

**Image Generation.**   We generate images from a relational scene description $R$ by sampling from the probability distribution $p(\boldsymbol{x}|R)$ using Langevin sampling on the energy function $E_\theta(\boldsymbol{x}|R)$ from random noise.

**Image Editing.**   To edit an image $\boldsymbol{x}'$, we utilize the same probability distribution $p(\boldsymbol{x}|R)$ and Langevin sampling on the energy function $E_\theta(\boldsymbol{x}|R)$ but initialize sampling from the image we wish to edit $\boldsymbol{x}'$ instead of random noise.

**Relational Scene Understanding.**   We may further utilize the energy function $E_\theta(\boldsymbol{x}|R)$ as a tool for relational scene understanding by noting that $p(\boldsymbol{x}|R) \propto e^{-E_\theta(\boldsymbol{x}|R)}$. The output values of the energy function can be used as a matching score of the generated/edited image and the given scene relational scene description $R$.

# 4   Experiment

We conduct empirical studies to answer the following questions: (1) Can we learn relational models that can generate and edit complex multi-object scenes when given relational scene descriptions with multiple composed scene relations? (2) Can we use our model to generalize to scenes that are never seen in training? (3) Can we understand the set of relations in a scene and infer semantically equivalent descriptions?

To answer these questions, we evaluate the proposed method and baselines on image generation, image editing, and image classification on two main datasets, *i.e.* CLEVR [19] and iGibson [41]. We also test the image generation performance of the proposed model and baselines on a real-world dataset *i.e.* Blocks [27].

## 4.1   Datasets

**CLEVR.** We use $50,000$ pairs of images and relational scene descriptions for training. Each image contains $1 \sim 5$ objects and each object consists of five different attributes, including color, shape, material, size, and its spatial relation to another object in the same image. There are $9$ types of colors, $4$ types of shapes, $3$ types of materials, $3$ types of sizes, and $6$ types of relations.

**iGibson.** On the iGibson dataset, we use 30,000 pairs of images and relational scene descriptions for training. Each image contains $1 \sim 3$ objects and each object consists of the same five different types of attributes as the CLEVR dataset. There are $6$ types of colors, $5$ types of shapes, $4$ types of materials, $2$ types of sizes, and $4$ types of relations. The objects are randomly placed in the scenes.

**Blocks.** On the real-world Blocks dataset, a number of 3,000 pairs of images and relational scene descriptions are used for training. Each image contains $1 \sim 4$ objects and each object differs in color. We only consider the "above" and "below" relations as objects are placed vertically in the form of towers.

In the training set, each image's relational scene description only contains one scene relation and objects are randomly placed in the scene. We generated three test subsets that contain relational scene descriptions with a different number of scene relations to test the generation ability of the proposed methods and baselines. The *1R* test subset is similar to the training set where each relational scene

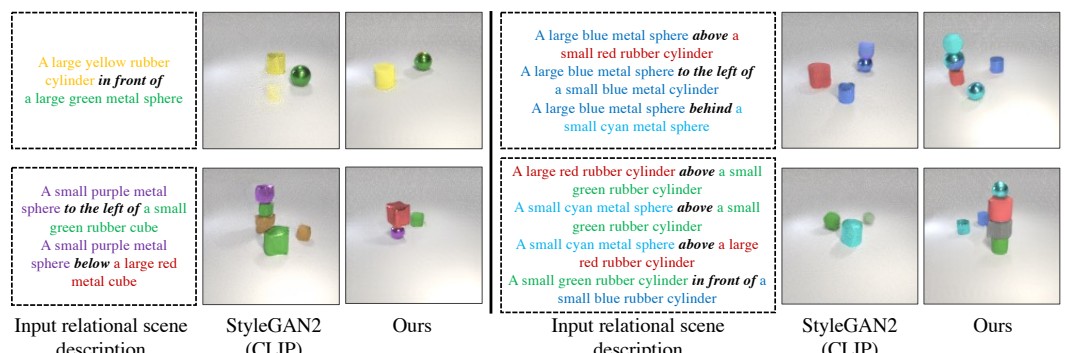

Figure 3: Image generation results on the CLEVR dataset. Image are generated based on $1 \sim 4$ relational descriptions. Note that the models are trained on a single relational description and the composed scene relations (2, 3, and 4 relational descriptions) are outside the training distribution.

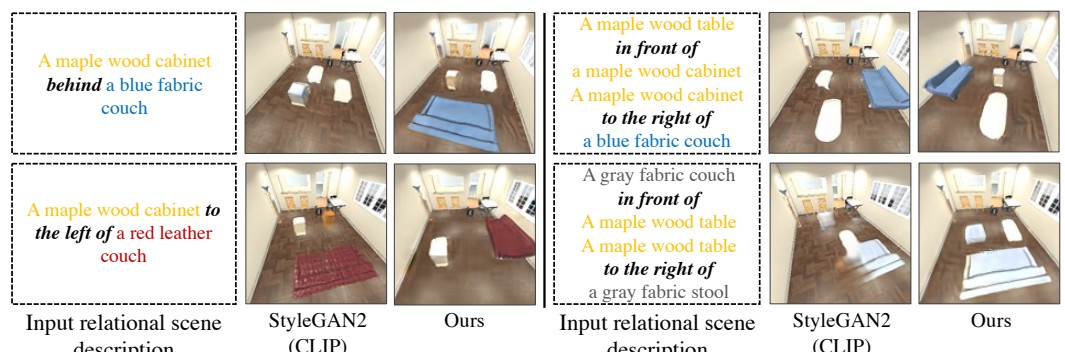

Figure 4: Image generation results on the iGibson dataset. Images are generated based on $1 \sim 2$ relational descriptions. Note that the two composed scene relations are outside the training distribution.

description contains one scene relation. The *2R* and *3R* test subsets have two and three scene relations in each relational scene description, respectively. Each test set has $5,000$ images with corresponding relational scene descriptions.

## 4.2 Baselines

We compare our method with two baseline approaches. The first baseline is StyleGAN2 [21], one of the state-of-the-art methods for unconditional image generation. To enable StyleGAN2 to generate images and edit images based on relational scene descriptions, we train a ResNet-18 classifier on top of it to predict the object attributes and their relations. Recently, CLIP [35] has achieved a substantial improvement on the text-image retrieval task by learning good text-image feature embeddings on large-scale datasets. Thus we design another baseline, StyleGAN2+CLIP, that combines the capabilities of both approaches. To do this, we encode relational scene descriptions into text embeddings using CLIP and condition StyleGAN2 on the embeddings to generate images. Please see Supplement Section E for more details of baselines.

## 4.3 Image Generation Results

Given a relational scene description, *e.g.* "a blue cube on top of a red sphere", we aim to generate images that contain corresponding objects and their relations as described in the given descriptions.

**Quantitative comparisons.** To evaluate the quality of generated images, we train a binary classifier to predict whether the generated image contains objects and their relations described in the given relational scene description.

Given a pair of an image and a relational scene description, we first feed the image to several convolutional layers to generate an image feature and then send the relational scene description to an embedding layer followed by several fully connected layers to generate a relational scene feature.

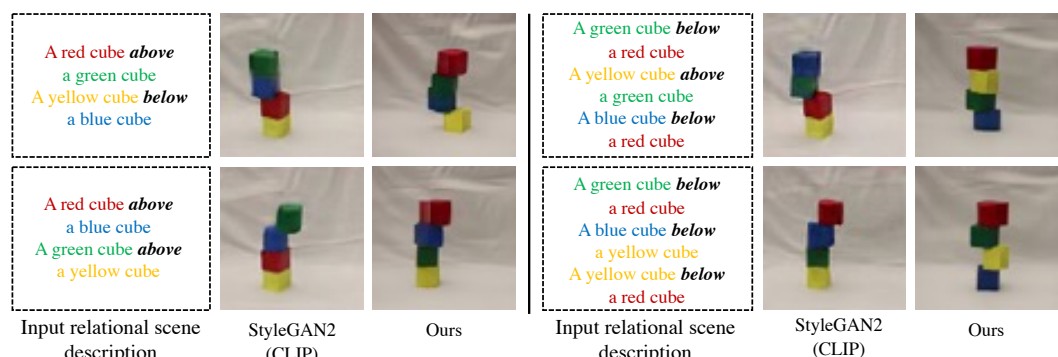

Figure 5: Image generation results on the Blocks dataset. Image are generated based the relational scene description. Note that the models are trained on a single relational scene description and the composed scene relations are outside the training distribution.

The image feature and relational scene feature are combined and then passed through several fully connected and finally a sigmoid function to predict whether the given image matches the relational scene description. The binary classifier is trained on real images from the training dataset. We train a classifier on each dataset and observe classification accuracy on real images to be close to 100%, indicating that the classifier is effective. During testing, we generate an image based on a relational scene description and send the generated image and the relational scene description to the classifier for prediction. For a fair comparison, we use the same classifier to evaluate images generated by all the approaches on each dataset.

The "Image Generation" column in Table 1 shows the classification results of different approaches on the CLEVR and iGibson datasets. On each dataset, we test each method on three test subsets, *i.e.* *1R*, *2R*, *3R*, and report their binary classification accuracies. Both variants of our proposed approach outperform StyleGAN2 and StyleGAN2 (CLIP), indicating that our method can generate images that contain the objects and their relations described in the relational scene descriptions. We find that our approach using the learned embedding, *i.e.* Ours (Learned Embed), achieves better performances on the CLEVR and iGibson datasets than the other variant using the CLIP embedding, *i.e.* Ours (CLIP).

StyleGAN2 and StyleGAN2 (CLIP) can perform well on the *1R* test subset. This is an easier test subset because the models are trained on images with a single scene relation and the models generate images based on a single relational scene description during testing as well. The *2R* and *3R* are more challenging test subsets because the models need to generate images conditioned on relational scene descriptions of multiple scene relations. Our models outperform the baselines by a large margin, indicating the proposed approach has a better generalization ability and can compose multiple relations that are never seen during training.

**Human evaluation results.** To further evaluate the performance of the proposed method on image generation, we conduct a user study to ask humans to evaluate whether the generated images match the given input scene description. We compare the correctness of the object relations in the generated images and the input language of our proposed model, *i.e.* "Ours (Learned Embed)", and "StyleGAN2 (CLIP)". Given a language description, we generate an image using "Ours (Learned Embed)" and "StyleGAN2 (CLIP)". We shuffle these two generated images and ask the workers to tell which image has better quality and the object relations match the input language description. We tested 300 examples in total, including 100 examples with 1 sentence relational description (1R), 100 examples with 2 sentence relational descriptions (2R), and 100 examples with 3 sentence relational descriptions (3R). There are 32 workers involved in this human experiment.

The workers think that there are 87%, 86%, and 91% of generated examples that "Ours (Learned Embed)" is better than "StyleGAN2 (CLIP)" for *1R*, *2R*, and *3R* respectively. The human experiment shows that our proposed method is better than "StyleGAN2 (CLIP)". The conclusion is coherent with our binary classification evaluation results.

**Qualitative comparisons.** The image generation results on CLEVR, iGibson and Block scenes are shown in Figure 3, 4, and 5, respectively. We show examples of generated images conditioned on relational scene descriptions of different number of scene relations. Our method generates images that are consistent with the relational scene descriptions. Note that both the proposed method and

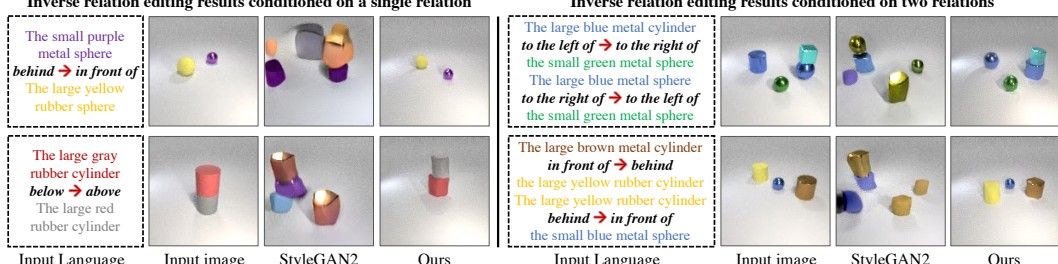

Figure 6: Image editing results on the CLEVR dataset. **Left**: image editing results based on a single relational scene description. **Right**: image editing results based on two composed relational scene descriptions. Note that the composed scene relations in the right part are outside the training distribution and our approach can still edit the images accurately.

the baselines are trained on images that only contain a relational scene description of a single scene relation describing the visual relationship between two objects in each image. We find that our approach can still generalize well when composing more visual relations. Taking the upper right figure in Figure 3 as an example, a relational scene description of multiple scene relations, *i.e.* "A large blue metal sphere above a small red rubber cylinder. A large blue metal sphere to the left of a small blue metal cylinder · · · ", is never seen during training. "StyleGAN2 (CLIP)" generates wrong objects and scene relations that are different from the scene descriptions. In contrast, our method has the ability to generalize to novel relational scenes.

### 4.4  Image Editing Results

Given an input image, we aim to edit this image based on relational scene descriptions, such as "put a red cube in front of the blue cylinder".

**Quantitative comparisons.** Similar to the image generation, we use a classifier to predict whether the image after editing contains the objects and their relations described in the relational scene description. For the evaluation on each dataset, we use the same classifier for both image generation and image editing.

The "Image Editing" column in Table 1 shows the classification results of different approaches on the CLEVR and iGibson datasets. Both variants of our proposed approach, *i.e.* "Ours (CLIP)" and "Ours (Learned Embed)" outperform the baselines, *i.e.* "StyleGAN2" and "StyleGAN2 (CLIP)", substantially. The good performance of our approach on the *2R* and *3R* test subsets shows that the proposed method has a good generalization ability to relational scene descriptions that are outside the training distribution. The images after editing based on relational scene descriptions can incorporate the described objects and their relations accurately.

**Qualitative comparisons.** We show image editing examples in Figure 6. The left part is image editing results conditioned on a single scene relation while the right part is conditioned on two scene relations. We show examples that edit images by inverting individual spatial relations between given two objects. Taking the first image in Figure 6 as an example, "the small purple metal sphere" is behind "the large yellow rubber sphere", after editing, our model can successfully put "the small purple metal sphere" in front of "the large yellow rubber sphere". Even for relational scene descriptions of two scene relations that are never seen during training, our model can edit images so that the selected objects are placed correctly.

### 4.5  Relational Understanding

We hypothesize the good generation performance of our proposed approach is due to our system's understanding of relations and ability to distinguish between different relational scene descriptions. In this section, we evaluate the relational understanding ability of our proposed method and baselines by comparing their image-to-text retrieval and semantic equivalence results.

**Image-to-text retrieval.** In Figure 7, we evaluate whether our proposed model can understand different relational scene descriptions by image-to-text retrieval. We create a test set that contains 240 pairs of images and relational scene descriptions. Given a query image, we compute the similarity of

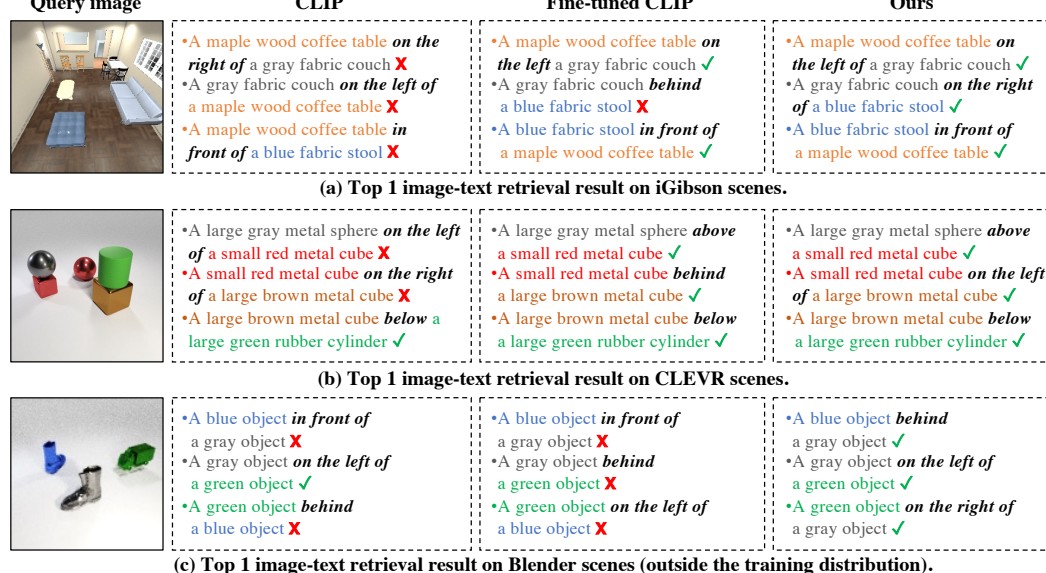

**(a) Top 1 image-text retrieval result on iGibson scenes.**

**(b) Top 1 image-text retrieval result on CLEVR scenes.**

**(c) Top 1 image-text retrieval result on Blender scenes (outside the training distribution).**

Figure 7: Image-to-text retrieval results. We compare the proposed approach with the pretrained CLIP and fine-tuned CLIP and show their top-1 retrieved relation description based on the given image query.

this image and each relational scene description in the gallery set. The top 1 retrieved relational scene description is shown in Figure 7. We compare our method with two baselines. We use the pre-trained CLIP model and test it on our dataset directly. "Fine-tuned CLIP" means the CLIP model is fine-tuned on our dataset. Even though CLIP has shown good performance on the general image-text retrieval task, we find that it cannot understand spatial relations well, while EBMs can retrieve all the ground truth descriptions.

We also find that our approach generalizes across datasets. In the bottom row of Figure 7, we conduct an additional image-to-text retrieval experiment on the Blender [4] scenes that are never seen during training. Our approach can still find the correct relational scene description for the query image.

**Can we understand semantically equivalent relational scene descriptions?** Given two relational scene descriptions describing the same image but in different ways, can our

Table 2: Quantitative evaluation of **semantic equivalence** on the CLEVR dataset.

| Model | Semantic Equivalence (%) | | |
|---|---|---|---|
| | 1R Acc | 2R Acc | 3R Acc |
| Classifier | 52.82 | 27.76 | 14.92 |
| CLIP | 37.02 | 14.40 | 5.52 |
| CLIP (Fine-tuned) | 60.02 | 35.38 | 20.9 |
| Ours (CLIP) | 70.68 | 50.48 | 38.06 |
| Ours (Learned Emb) | **74.76** | **57.76** | **44.86** |

approach understand that the descriptions are semantically similar or equivalent? To evaluate this, we create a test subset that contains $5,000$ images and each image has 3 different relational scene descriptions. There are two relational scene descriptions that match the image but describe the image in different ways, such as "a cabinet in front of a couch" and "a couch behind a cabinet". There is one further description that does not match the image. The relative score difference between the two ground truth relational scene descriptions should be smaller than the difference between one ground truth relational scene description and one wrong relational scene description.

We compare our approach with three baselines. For each model, given an image, if the difference between two semantically equivalent relational scene description is smaller than the difference between the semantically different ones, we will classify it as correct. We compute the percentage of correct predictions and show the results in Table 2. Our proposed method outperforms the baselines substantially, indicating that our EBMs can distinguish semantically equivalent relational scene descriptions and semantically nonequivalent relational scene descriptions. In Figure 8, We further show two examples generated by our approach on the iGibson and CLEVR datasets. The energy difference between the semantically equivalent relational scene description is smaller than the mismatching pairs.

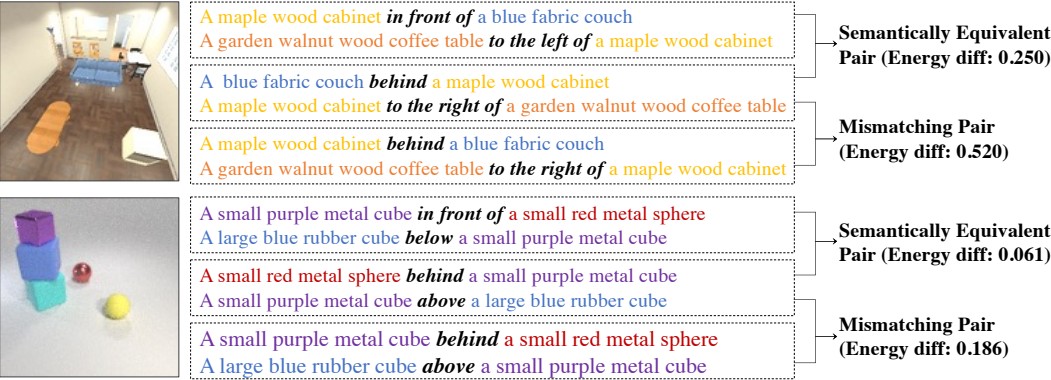

Figure 8: Examples of **semantic equivalence** on CLEVR and iGibson scenes. Given an input image, our approach is able to recognize whether the relational scene descriptions are semantically equivalent or not.

## 4.6 Zero-shot Generalization Across Datasets

We find that our method can generalize across datasets as shown in the third example in Figure 7. To quantitatively evaluate the generalization ability across datasets of the proposed method, we test the image-to-text retrieval accuracy on the Blender dataset. We render a new Blender dataset using objects including boots, toys, and trucks. Note that our model and baselines are trained on CLEVR and have never seen the Blender scenes during training.

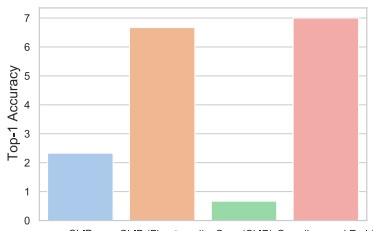

Figure 9: **Zero-shot generalization** on Blender scenes. Our approach with learned embedding outperforms other methods on image-to-text retrieval.

We generate a Blender test set that contains 300 pairs of images and relational scene descriptions. For each image, we do text retrieval on the 300 relational scene descriptions. The top 1 accuracy is shown in Figure 9. We compare our approaches with two baselines, *i.e.* CLIP and CLIP fine-tuned on the CLEVR dataset. We find the CLIP model and our approach using the CLIP embedding perform badly on the Blender dataset. This is because CLIP is not good at modeling relational scene description, as we have shown in Section 4.5. Our approach using the learned embedding outperforms other methods, indicating that our EBMs with a good embedding feature can generalize well even on unseen datasets, such as Blender.

## 5 Conclusion

In this paper, we demonstrate the potential usage for our model on compositional image generation, editing, and even generalization on unseen datasets given only relational scene descriptions. Our results provide evidence that EBMs are a useful class of models to study relational understanding.

One limitation of the current approach is that the evaluated datasets are simpler compared to the complex relational descriptions used in the real world. A good direction for future work would be to study how these models scale to complex datasets found in the real world. One particular interest could be measuring the zero-shot generalization capabilities of the proposed model.

Our system, as with all systems based on neural networks, is susceptible to dataset biases. The learned relations will exhibit biases if the training datasets are biased. We must take balanced and fair datasets if we develop our model to solve real-world problems, as otherwise, it could inadvertently worsen existing societal prejudices and biases.

**Acknowledgements.** Shuang Li is supported by Raytheon BBN Technologies Corp. under the project Symbiant (reg. no. 030256-00001 90113) and Mitsubishi Electric Research Laboratory (MERL) under the project Generative Models For Annotated Video. Yilun Du is supported by NSF graduate research fellowship and in part by ONR MURI N00014-18-1-2846 and IBM Thomas J. Watson Research Center CW3031624.

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
