# Learning to Compose Visual Relations
# Supplementary Material

**Nan Liu** *
University of Michigan
liunan@umich.edu

**Shuang Li** *
MIT CSAIL
lishuang@mit.edu

**Yilun Du** *
MIT CSAIL
yilundu@mit.edu

**Joshua B. Tenenbaum**
MIT CSAIL, BCS, CBMM
jbt@mit.edu

**Antonio Torralba**
MIT CSAIL
torralba@mit.edu

In this supplementary material, we first present qualitative and quantitative comparisons with additional baseline approaches in Section A. We then show the experiment results on the more complex real world datasets in Section B and additional dataset details in Section C. More details of our proposed approach and baselines are shown in Section D and Section E, respectively. We provide the model architecture details of various approaches and their implementation details in Section F and Section G, respectively. Finally, we show the psuedocode for our algorithm in Section H.

## A    Additional Results

**Comparison with more baseline approaches.**    We provide more results of additional baselines in Table 1. We use the same evaluation metrics as in Table 1 of the main paper. The details of baselines are described in Section E of this supplement. As shown in Table 1 of this supplement, our approach achieves the highest accuracy among all the methods for both image generation and image editing. In Table 1, as noted in the main paper, directly encoding a relational scene description such as "a large blue rubber cube to the left of a small red metal cube" utilizing CLIP to train an EBM ("EBM (CLIP) (Full Sentence)") performs much worse than the proposed method "Ours (CLIP)" and "Ours (Learned Embed)".

**Additional evaluation metric.**    In addition to comparing the binary classification accuracy of different methods as we used in Table 1 of the main paper, we provide an additional evaluation metric for image generation. We investigate the performance of utilizing the graph-based relational similarity metric proposed by [2] for image generation. A graph-based relational similarity score is used to test the correct placement of objects, without requiring the model to draw the objects exactly in the same locations as the ground truth. Such a metric can construct scene graphs for both the generated and ground truth images without telling the model to precisely draw objects at the exact locations. However, it heavily relies on the pre-trained object detector and localizer. The pre-trained object detector or localizer could generate false predictions on both real images and generated images, especially when the generated images are out of the training distribution.

As the evaluation metric used in [2] focuses more on the local matching while our binary classification focuses on the global matching, in this supplement, we further report the results for two baselines and our approach using the evaluation metric proposed by [2]. The image generation results on the CLEVR dataset are listed in Table 2. The conclusion obtained by using this new metric is coherent with using our binary classification metric (Table 1 of the main paper): our proposed method outperforms the baselines.

---

*indicates equal contribution

35th Conference on Neural Information Processing Systems (NeurIPS 2021).

Table 1: Evaluation of the accuracy of object relations in the generated images or edited images on the CLEVR and iGibson datasets. We compare our method with baselines on three test sets, *i.e.* *1R*, *2R*, and *3R*. In Table 1 of the main paper, we had two baselines, *i.e.* StyleGAN2 and StyleGAN2 (CLIP). Here we add another 3 baselines, *i.e.* Scene Graph GAN [3], EBM (CLIP) (Full Sentence), and StyleGAN2 (CLIP) (Multi-Relations), for comparison.

| Dataset | Model | Image Generation (%) | | |
|---------|-------|--------|--------|--------|
| | | 1R Acc | 2R Acc | 3R Acc |
| CLEVR | StyleGAN2 | 10.68 | 2.46 | 0.54 |
| | StyleGAN2 (CLIP) | 65.98 | 9.56 | 1.78 |
| | StyleGAN2 (CLIP) (Multi-Relations) | 66.62 | 9.60 | 1.68 |
| | Scene Graph GAN | 83.72 | 14.18 | 4.48 |
| | EBM (CLIP) (Full Sentence) | 16.24 | 1.46 | 0.11 |
| | Ours (CLIP) | 94.79 | 48.42 | 18.00 |
| | Ours (Learned Embed) | **97.79** | **69.55** | **37.60** |
| iGibson | StyleGAN2 | 12.46 | 2.24 | 0.60 |
| | StyleGAN2 (CLIP) | 49.20 | 17.06 | 5.10 |
| | StyleGAN2 (CLIP) (Multi-Relations) | 36.94 | 13.42 | 6.86 |
| | Scene Graph GAN | 54.64 | 0.02 | 0.00 |
| | EBM (CLIP) (Full Sentence) | 36.92 | 12.72 | 4.63 |
| | Ours (CLIP) | 74.02 | 43.04 | **19.59** |
| | Ours (Learned Embed) | **78.27** | **45.03** | 19.39 |

Table 2: Comparison of different methods on the CLEVR dataset. The accuracy of **graph-based relational similarity** proposed by [2] is reported.

| Model | Relational Similarity (%) | | |
|-------|--------|--------|--------|
| | 1R Acc | 2R Acc | 3R Acc |
| StyleGAN2 | 22.37 | 19.75 | 17.13 |
| StyleGAN2 (CLIP) | 37.50 | 28.62 | 28.75 |
| Ours (Learned Emb) | **50.77** | **36.87** | **42.50** |

**Additional qualitative results.** We show more qualitative results of image generation in Figure 1 and Figure 2. Our approach can generate images with correct relations, and can even generalize to relational scene descriptions that are out of the training distribution.

# B  Image Generation Results on Real World Datasets

In terms of image generation on real scenes, we train and evaluate our model on two real-world datasets, the Blocks dataset [7] and the Visual Genome dataset [6].

The Blocks dataset is from [7] and we train our model using the object relations, e.g. "above" and "below". We show the images generated conditioned on two relational descriptions and three relational descriptions in Figure 3.

For the Visual Genome dataset [6], we train our models on a subset that consists of common objects and relations for computational efficiency. As shown in Figure 4, we find that the CLIP text encoder performs better, as it has seen large-scale image-text pairs that cover a wide range of relations, attributes and objects.

Our approach is able to generate images (objects and their relations) matching the given language descriptions on the real-world Blocks dataset and the Visual Genome dataset. The quality of generated images on the Blocks dataset is great. However, the quality of results on the Visual Genome dataset is a bit worse. We believe that the generation quality could be further improved.

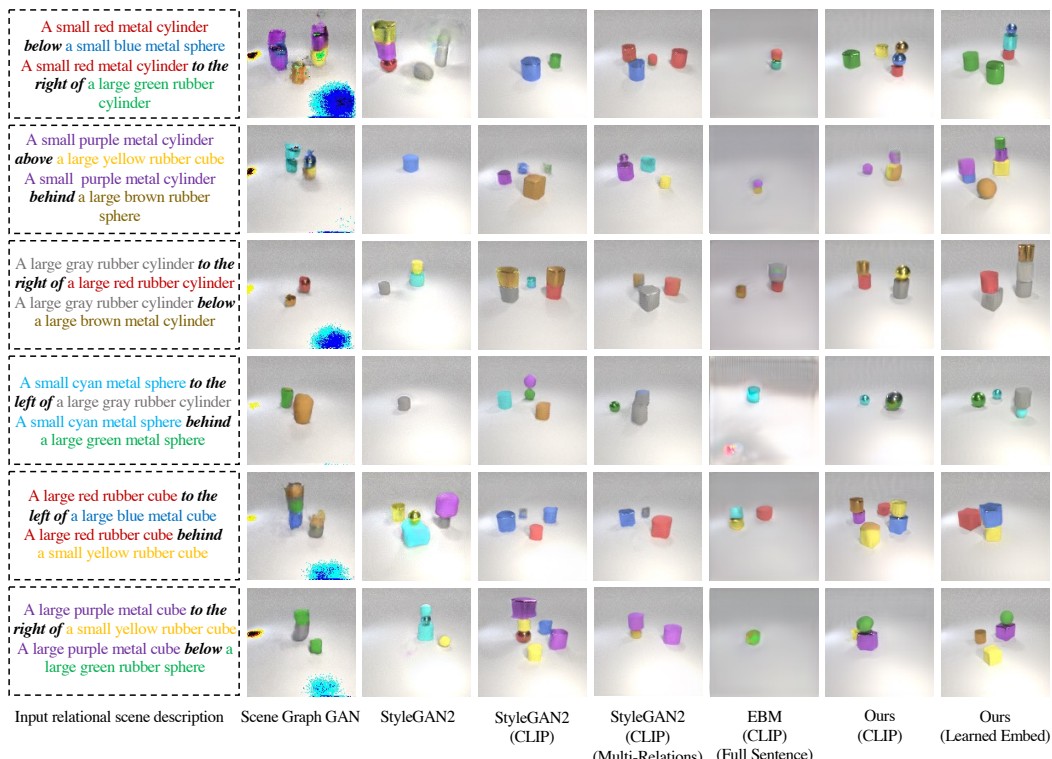

| Input relational scene description | Scene Graph GAN | StyleGAN2 | StyleGAN2 (CLIP) | StyleGAN2 (CLIP) (Multi-Relations) | EBM (CLIP) (Full Sentence) | Ours (CLIP) | Ours (Learned Embed) |

Figure 1: Image generation results on the CLEVR dataset. Image are generated based on 2 relational descriptions. Note that the models are trained on a single relational description and the two composed scene relations are outside the training distribution. Our approaches "Ours (CLIP)" and "Ours (Learned Embed)" are able to generate images accurately based on the input scene descriptions.

## C  Datasets Details

**CLEVR.** On the CLEVR dataset, each image contains $1 \sim 5$ objects and each object consists of five different attributes, including color, shape, material, size, and its relation to another object in the same image. There are 9 types of colors, 4 types of shapes, 3 types of materials, 3 types of sizes, and 6 types of relations. The objects are randomly placed in the scenes.

**iGibson.** On the iGibson dataset, each image contains $1 \sim 3$ objects and each object consists of the same five different types of attributes as the CLEVR dataset. There are 6 types of colors, 5 types of shapes, 4 types of materials, 2 types of sizes, and 4 types of relations. The objects are randomly placed in the scenes.

**Blocks.** On the real-world Blocks dataset, each image contains $1 \sim 4$ cubes and each cube only differs in color. Objects in the images are placed vertically in the form of towers.

There are 50,000, 30,000 and 3,000 training images on the CLEVR, iGibson and Blocks datasets, respectively, and 5,000 testing images on both the CLEVR and iGibson datasets. We test the zero-shot generalization across datasets using the blender data. There are three types of objects, including *trucks*, *toys*, and *boots*. We generated 5,000 testing images with each image contains $1 \sim 3$ objects for the Blocks dataset. There is no overlap between the training and testing data on each dataset.

## D  Details of Our Approaches

**Ours (CLIP).**  In our EBM setting, we use the pre-trained CLIP model to encode objects and a learned embedding layer to encode their relations. Taking the scene description of "a large blue rubber cube to the left of a small red metal cube" as an example, we use the pre-trained CLIP model to encode the two objects seperately, *i.e.* $o^1$ for "a large blue rubber cube" and $o^2$ for "a small red metal cube". We then use an embedding layer to encode their relation, *i.e.* $r'$ for "to the left". The features of the first and second objects and their relations are concatenated and used as the feature of

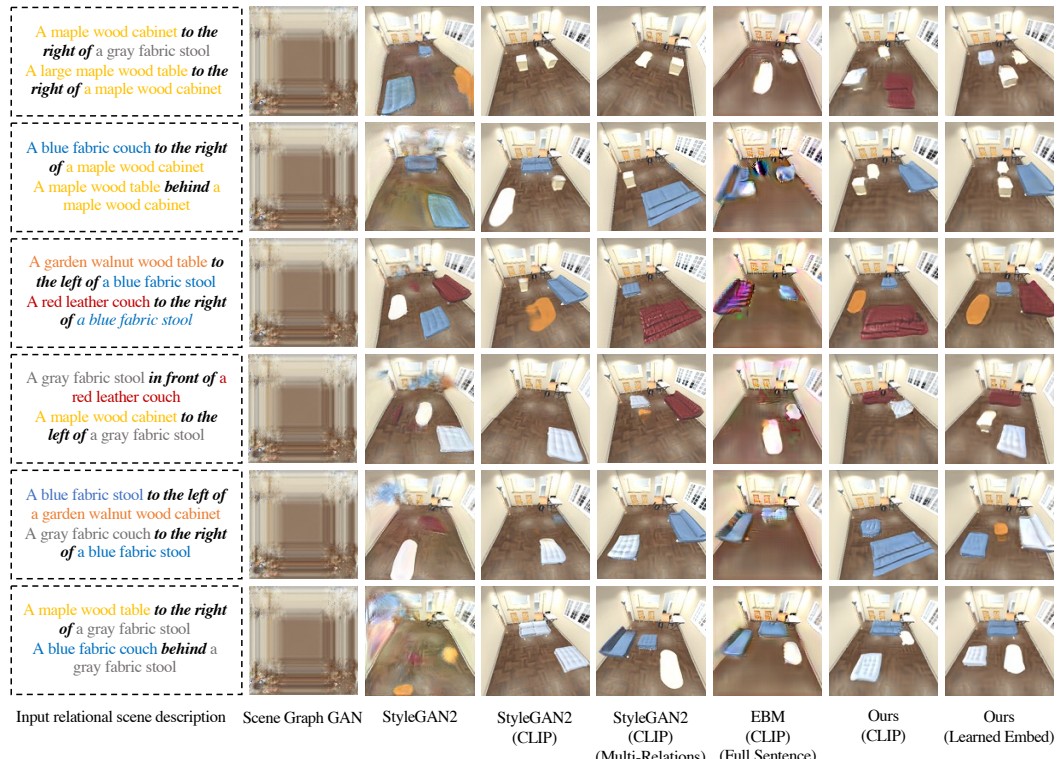

Figure 2: Image generation results on the iGibson dataset. Image are generated based on 2 relational descriptions. Note that the models are trained on a single relational description and the two composed scene relations are outside the training distribution. Our approaches "Ours (CLIP)" and "Ours (Learned Embed)" are able to generate images accurately based on the input scene descriptions.

the relational scene description which is further send to the relational energy functions $E_\theta$ for image generation or image editing.

**Ours (Learned Embed).** Different from "Ours (CLIP)", "Ours (Learned Embed)" uses the learned embedding layers for both objects and their relations. To encode an object, we use 6 different embedding layers to learn its color, size, material, shape, relation and position, seperately. The embedded features of objects and their relations are concatenated and used as the feature of the relational scene description which is further sent to the relational energy functions $E_\theta$ for image generation or image editing.

## E  Details of Baselines

**StyleGAN2.** In Section 4.2 of the main paper, we used the unconditional StyleGAN2 [4] as one of the baselines. We train the unconditional StyleGAN2 and the ResNet-18 classifier separately on each dataset. For training, we use the default setting provided by [4]. To generate an image with respect to a particular relation, we optimize the underlying latent code to minimize the loss from the classifier.

**StyleGAN2 (CLIP).** StyleGAN2 (CLIP) is the same as StyleGAN2 except that StyleGAN2 (CLIP) uses the text encoder of the CLIP model [8] to encode relational scene descriptions. We follow the same configuration as the StyleGAN2 to train StyleGAN2 (CLIP).

**StyleGAN2 (CLIP) (Multi-Relations).** StyleGAN2 (CLIP) (Multi-Relations) has the same model architecture as StyleGAN2 (CLIP) but is trained with more scene relations. In StyleGAN2 (CLIP), we only use a single scene relation during training while StyleGAN2 (CLIP) (Multi-Relations) uses $1 \sim 3$ scene relations.

**Scene Graph GAN.** We apply the models from [3] and utilize the extracted scene graphs as input to train a conditional StyleGAN2. As there is no object bounding boxes available in our setting, we

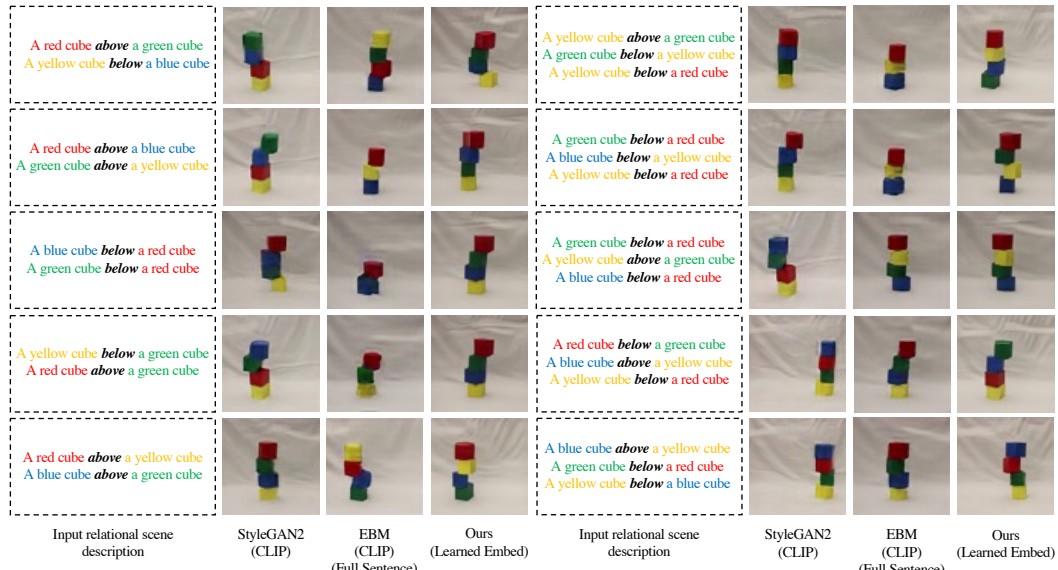

Figure 3: Image generation results on the Blocks dataset. Image are generated based on 2 or 3 relational descriptions. Note that the models are trained on a single relational description and the composed scene relations (2 or 3 relational descriptions) are outside the training distribution. Our approach "Ours (Learned Embed)" is able to generate images accurately based on the input scene descriptions.

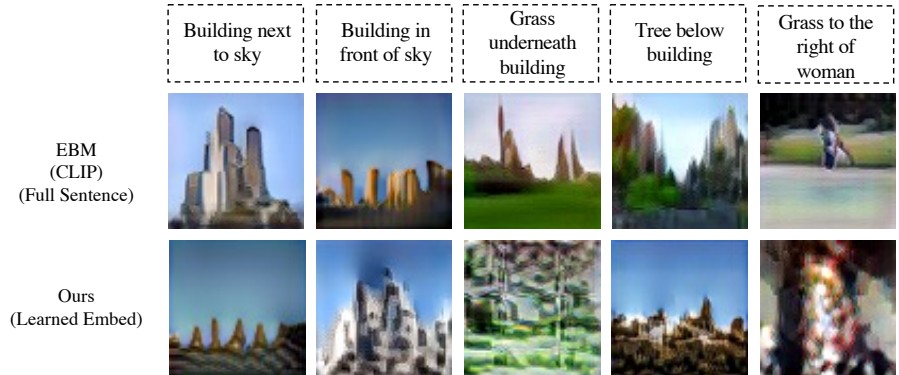

Figure 4: Image generation results on the Visual Genome dataset. "EBM (CLIP) (Full Sentence)" performs better than our approach "Ours (Learned Emb)" on generating more complex natural images because pretrained CLIP text encoder has seen large-scale image-text pairs that cover a wide range of relations and objects.

set the input bounding box to be the whole image frame and our input scene graphs only consist of two objects and their relation.

**EBM (CLIP) (Full Sentence).** In this setting, we use the text encoder of CLIP to encode every word in the relational scene descriptions. Such a holistic encoder has a bad performance as shown in Table 1, Figure 1, Figure 2 and Figure 3.

## F   Model Architecture Details

We follow the implementation of EBMs from [1] in our experiments. Similar to [1], we use the multi-scale model architecture to compute energies as shown in Table 3. Each model generates an energy value and the final energy $E_\theta(\mathbf{x})$ is the sum of energies from all the models listed in Table 3. Given relational scene descriptions, we generate or edit images based on the final energy.

Table 3: We use the multi-scale model architecture to compute energies as in [1].

| |
| --- |
| 3x3 Conv2d 128 |
| CondResBlock 128 |
| CondResBlock Down 128 |
| CondResBlock 128 |
| CondResBlock Down 256 |
| Self-Attention 256 |
| CondResBlock 256 |
| CondResBlock Down 256 |
| CondResBlock 512 |
| CondResBlock Down 512 |
| Global Mean Pooling |
| Dense → 1 |

| |
| --- |
| 3x3 Conv2d 128 |
| CondResBlock 128 |
| CondResBlock Down 128 |
| CondResBlock 128 |
| CondResBlock Down 128 |
| Self-Attention 256 |
| CondResBlock 256 |
| CondResBlock Down 256 |
| Global Mean Pooling |
| Dense → 1 |

| |
| --- |
| 3x3 Conv2d 128 |
| CondResBlock 128 |
| CondResBlock Down 128 |
| Self-Attention 128 |
| CondResBlock 128 |
| CondResBlock Down 128 |
| Global Mean Pooling |
| Dense → 1 |

# G   Implementation Details

**StyleGAN2.**   It takes 2 days to train the StyleGAN2 model and 2 hours to train the classifier using a single Tesla 32GB GPU on each dataset. We use the Adam optimizer [5] with $\beta_1 = 0$, $\beta_2 = 0.99$, and $\epsilon = 10^{-8}$ to train the model.

**StyleGAN2 (CLIP).**   For StyleGAN2 (CLIP) and StyleGAN2 (CLIP) (Multi-Relations), it takes around 2 days to train each of them on each dataset using a single Tesla 32GB GPU. We use the Adam optimizer [5] with $\beta_1 = 0$, $\beta_2 = 0.99$, and $\epsilon = 10^{-8}$ to train them.

**Scene Graph GAN.**   We train the model on each dataset with the default training configuration provided in the codebase from [3] for 2 days using a single Tesla 32GB GPU. We use the Adam optimizer [5] with $\beta_1 = 0.9$, $\beta_2 = 0.999$, and $\epsilon = 10^{-4}$ to train the model.

**EBMs (*i.e.*, Ours (CLIP), Ours (Learned Embed), EBM (CLIP) (Full Sentence)).**   In our experiments, we use the same setting to train models using EBMs, *i.e.*, Ours (CLIP), Ours (Learned Embed), and EBM (CLIP) (Full Sentence), for fair comparison. We use the Adam optimizer [5] with learning rates of $10^{-4}$ and $2 \times 10^{-4}$ on the CLEVR and iGibson datasets, respectively. For MCMC sampling, we use a step size of 300 on the CLEVR dataset, 750 on the iGibson dataset and 300 on the Blocks dataset. On each dataset, the model is trained for 3 days on a single Tesla 32GB GPU.

To generate images at test time, we initialize an image sample from random noise. We then iteratively apply data augmentation on the image sample followed by 20 steps of Langevin sampling. To generate the final image, we run 80 additional steps of Langevin sampling on the image sample.

To edit images at test time, we run 80 steps of Langevin sampling on the image to edit. The step size of Langevin sampling is inversely proportional to the number of scene relations, *i.e.* more scene relations leads to a lower Langevin sampling step size.

# H   Algorithms

We provide the algorithms of the proposed method, including training, image generation, image editing, and image-to-text retrieval, in Algorithm 1, 2, 3 and 4, respectively.

---

**Algorithm 1** Conditional EBM training algorithm

---

**Input:** data dist $p_D(\boldsymbol{x})$, relational scene descriptions $R_D(\boldsymbol{r})$, step size $\lambda$, number of steps $M$, data augmentation $D(\cdot)$, stop gradient operator $\Omega(\cdot)$, EBM $E_\theta(\cdot)$, Encoder $\mathrm{Enc}(\cdot)$
$\mathcal{B} \leftarrow \varnothing$
**while** not converged **do**
    $\boldsymbol{x}_i^+ \sim p_D$
    $R_i \sim R_D$
    $\tilde{\boldsymbol{x}}_i^0 \sim \mathcal{B}$ with 99.9% probability and $\mathcal{U}$ otherwise
    $X \sim \mathcal{B}$ for nearest neighbor entropy calculation

    ▷ *Split a relational scene description into individual scene relations:*
    $\{\boldsymbol{r}_1, \ldots \boldsymbol{r}_K\} \leftarrow R_i$

    ▷ *Apply data augmentation to sample:*
    $\tilde{\boldsymbol{x}}_i^0 = D(\tilde{\boldsymbol{x}}_i^0)$

    ▷ *Generate sample using Langevin dynamics:*
    **for** sample step $m = 1$ to $M$ **do**
        $\tilde{\boldsymbol{x}}_i^{m-1} = \Omega(\tilde{\boldsymbol{x}}_i^{m-1})$
        $\tilde{\boldsymbol{x}}^m \leftarrow \tilde{\boldsymbol{x}}^{m-1} - \nabla_{\boldsymbol{x}} \sum_{k=1}^{K} \frac{\lambda}{K} \cdot E_\theta(\tilde{\boldsymbol{x}}^{m-1} \mid \mathrm{Enc}(\boldsymbol{r}_k)) + \omega^m, \ \ \omega^m \sim \mathcal{N}(0, \sigma)$
    **end for**

    ▷ *Generate two variants of $\boldsymbol{x}^-$ with and without gradient propagation:*
    $\boldsymbol{x}_i^- = \Omega(\tilde{\boldsymbol{x}}_i^m)$
    $\hat{\boldsymbol{x}}_i^- = \tilde{\boldsymbol{x}}_i^m$

    ▷ *Optimize objective $\mathcal{L}_{CD} + \mathcal{L}_{KL}$ wrt $\theta$:*
    $\mathcal{L}_{\mathrm{CD}} = \frac{1}{N} \sum_i \sum_{k=1}^{K} (E_\theta(\boldsymbol{x}_i^+ \mid \mathrm{Enc}(\boldsymbol{r}_k)) - E_\theta(\boldsymbol{x}_i^- \mid \mathrm{Enc}(\boldsymbol{r}_k)))$
    $\mathcal{L}_{\mathrm{KL}} = \sum_{k=1}^{K} E_{\Omega(\theta)}(\hat{\boldsymbol{x}}_i^- \mid \mathrm{Enc}(\boldsymbol{r}_k)) - \log(NN(\hat{\boldsymbol{x}}_i^-, X))$

    ▷ *Optimize objective $\mathcal{L}_{CD} + \mathcal{L}_{KL}$ wrt $\theta$:*
    $\Delta\theta \leftarrow \nabla_\theta(\mathcal{L}_{\mathrm{CD}} + \mathcal{L}_{\mathrm{KL}})$
    Update $\theta$ based on $\Delta\theta$ using Adam optimizer

    ▷ *Update replay buffer $\mathcal{B}$*
    $\mathcal{B} \leftarrow \mathcal{B} \cup \tilde{\boldsymbol{x}}_i^-$
**end while**

---

---

**Algorithm 2** Image generation during testing

---

**Input:** Relational scene description $R_i$, number of data augmentation applications $N$, step size $\lambda$, number of steps $M$, data augmentation $D(\cdot)$, EBM $E_\theta(\cdot)$, Encoder $\mathrm{Enc}(\cdot)$
$\tilde{\boldsymbol{x}}^0 \sim \mathcal{U}$

▷ *Split a relational scene description into individual scene relations:*
$\{\boldsymbol{r}_1, \ldots \boldsymbol{r}_K\} \leftarrow R_i$

▷ *Generate samples through N iterative steps of data augmentation/Langevin dynamics:*
**for** sample step $n = 1$ to $N$ **do**
    ▷ *Apply data augmentation to samples:*
    $\tilde{\boldsymbol{x}}^0 = D(\tilde{\boldsymbol{x}}_i^0)$

    ▷ *Run M steps of Langevin dynamics:*
    **for** sample step $m = 1$ to $M$ **do**
        $\tilde{\boldsymbol{x}}^m \leftarrow \tilde{\boldsymbol{x}}^{m-1} - \sum_{k=1}^{K} \frac{\lambda}{K} \cdot \nabla_{\boldsymbol{x}} E_\theta(\tilde{\boldsymbol{x}}^{m-1} \mid \mathrm{Enc}(\boldsymbol{r}_k)) + \omega^m, \ \ \omega^m \sim \mathcal{N}(0, \sigma)$
    **end for**

    ▷ *Iteratively refine samples:*
    $\tilde{\boldsymbol{x}}^0 = \tilde{\boldsymbol{x}}^m$
**end for**

▷ *Run $S = 80$ steps of Langevin dynamics:*
**for** sample step $s = 1$ to $S$ **do**
    $\tilde{\boldsymbol{x}}^s \leftarrow \tilde{\boldsymbol{x}}^{s-1} - \sum_{k=1}^{K} \frac{\lambda}{K} \cdot \nabla_{\boldsymbol{x}} E_\theta(\tilde{\boldsymbol{x}}^{s-1} \mid \mathrm{Enc}(\boldsymbol{r}_k)) + \omega^s, \ \ \omega^s \sim \mathcal{N}(0, \sigma)$
**end for**

$\tilde{\boldsymbol{x}}^0 = \tilde{\boldsymbol{x}}^s$

▷ *Final output:*
$\boldsymbol{x} = \tilde{\boldsymbol{x}}^0$

---

---

**Algorithm 3** Image editing during testing

---

**Input:** input image $\tilde{\boldsymbol{x}}^0$, relational scene description $R$, number of data augmentation applications $N$, step size $\lambda$, number of steps $M$, data augmentation $D(\cdot)$, EBM $E_\theta(\cdot)$ Encoder Enc$(\cdot)$

▷ *Split a relational scene description into individual scene relations:*
$\{\boldsymbol{r}_1, \ldots \boldsymbol{r}_K\} \leftarrow R$

▷ *Generate samples through $N$ iterative steps of data augmentation/Langevin dynamics:*
**for** sample step $n = 1$ to $N$ **do**
   ▷ *Run $M$ steps of Langevin dynamics:*
   **for** sample step $m = 1$ to $M$ **do**
      $\tilde{\boldsymbol{x}}^m \leftarrow \tilde{\boldsymbol{x}}^{m-1} - \sum_{k=1}^{K} \frac{\lambda}{K} \cdot \nabla_{\boldsymbol{x}} E_\theta(\tilde{\boldsymbol{x}}^{m-1} \mid \text{Enc}(\boldsymbol{r}_k)) + \omega^m, \;\; \omega^m \sim \mathcal{N}(0, \sigma)$
   **end for**

   ▷ *Iteratively refine samples:*
   $\tilde{\boldsymbol{x}}^0 = \tilde{\boldsymbol{x}}^m$
**end for**

▷ *Final output:*
$\boldsymbol{x} = \tilde{\boldsymbol{x}}^0$

---

---

**Algorithm 4** Image-to-text retrieval during testing

---

**Input:** input image $\boldsymbol{x}$, relational scene descriptions $\{R_1, \ldots, R_n\}$, EBM $E_\theta(\cdot)$, Encoder Enc$(\cdot)$, output energy list $\mathcal{O}$, caption prediction $\mathcal{C}$
$\mathcal{O} \leftarrow []$

▷ *Generate image-caption matching energies iteratively*
**for** number of scene relations descriptions $i = 1$ to $n$ **do**
   ▷ *Split a relational scene description into individual scene relations:*
   $\{\boldsymbol{r}_1, \ldots \boldsymbol{r}_K\} \leftarrow R_i$
   $\boldsymbol{e}_i = \sum_{k=1}^{K} E_\theta(\boldsymbol{x} \mid \text{Enc}(\boldsymbol{r}_k))$

   ▷ *output energy list $\mathcal{O}$*
   $\mathcal{O}.append(\boldsymbol{e}_i)$
**end for**

▷ *Final output:*
$\mathcal{C} = \arg\min \mathcal{O}$

---