# OpenReview forum: "Learning to Compose Visual Relations"
_NeurIPS.cc/2021/Conference — NeurIPS 2021 Spotlight_

### Official Review · Reviewer_TEjR · 2021-07-12

**Rating:** 7
**Confidence:** 5

**Summary:**

This paper demonstrates the potential usage of our models across tasks of compositional multi-object relational image generation, editing, and even generalization on unseen datasets. The proposed model represents each relation as an unnormalized density (an energy-based model), enabling it to compose separate relations in a factorized manner. The experimental results significantly outperform baseline approaches and even can generalize to a previously unseen relation description.


**Ethical Concerns:**

No.

**Limitations And Societal Impact:**

Yes.

**Main Review:**

**Strengths:**
1) Overall, this paper is well written, and the technical details are easy to follow.
2) The main idea of a framework that factorizes and composes separate object relations is novel.
3) I found it very compelling that the approach can generalize to a previously unseen relation description.


**Weaknesses:**

1) **Motivation.** The authors mentioned in line 28, ‘’We posit that an issue with these models is that the language encoder is not compositional with respect to the underlying relations in an image, thus they fail to consider every single relation that describes a scene.’’ I would like to see evidence for that (citations or experiments that prove that this is the case). In general, I assume that there are language encoders that could contain enough compositionality, such as GPT3, and etc.

2) **Related Work.** As the authors mentioned in the related work section, the proposed idea has similarities with scene graphs. In the scene graph-to-image task, we model a set of triplets (<subject, relation, object>) as a structure and generate from it an image/video. I would like to mention [1], which model set of triplets (with the notion of time) to generate videos given a structure. Additionally, [2] generates videos given a set of descriptions the same as the authors proposed. I believe they could be mentioned.

	[1] Compositional Video Synthesis with Action Graphs, ICML 2021.

	[2] Concept Grounding with Modular Action-Capsules in Semantic Video Prediction.

3) **Human eval.** In my opinion, Tables 1 and 2 are missing human evaluation. Since generation and semantic equivalence are challenging to examine with current metrics, I suggest performing human analysis.

4) **Relation Descriptions Vs. Scene Graphs (SGs).** I think it is interesting to understand the difference between relation descriptions-to-image and scene graphs-to-image tasks. Of course, both tasks aim to do the same, but it is still unclear which is a more appropriate direction for image generation, image editing, and generalization capabilities. Thus, I would expect to see a comparison of the proposed model and other existing SG-to-image methods.


I raised some of my concerns above. Overall, I think it is an interesting paper that can be valuable to the community. I am open to the authors' feedback and other reviewers' opinions.



**Time Spent Reviewing:**

6-8

---

> ### Author Response · Authors · 2021-08-10
> **Author Response to Reviewer TEjR**
>
> **High-level response**
>
> Thank you for your detailed comments and feedback; we appreciate the time and attention you spent on reviewing our paper. We have addressed concerns about additional baselines, motivation clarifications, and have added human evaluation. Please feel free to let us know if you have more questions about the paper. We will try our best to address your concerns.
>
> ---------------------
> **Q1: Motivation. The authors mentioned in line 28, ‘’We posit that an issue with these models is that the language encoder is not compositional with respect to the underlying relations in an image, thus they fail to consider every single relation that describes a scene.’’ I would like to see evidence for that (citations or experiments that prove that this is the case). In general, I assume that there are language encoders that could contain enough compositionality, such as GPT3, and etc.**
>
> One source of evidence towards a language model lacking compositionality is the DALL·E model from OPEN-AI. DALL·E is a 12-billion parameter version of GPT-3 trained to generate images from text descriptions, using a dataset of text–image pairs. In their webpage (https://openai.com/blog/dall-e/), they have mentioned that *“We find that DALL·E correctly responds to some types of relative positions, but not others. The choices "sitting on” and "standing in front of" sometimes appear to work, "sitting below", "standing behind", "standing left of", and "standing right of" do not. DALL·E also has a lower success rate when asked to draw a large object sitting on top of a smaller one, when compared to the other way around”*. We also suggest the reviewer try to change the input text in the "Drawing Multiple Objects" section in the DALL·E webpage. We found that the generated images have a very low success rate to match the input text, indicating the underlying lack of compositionality in the language encoder.
>
> Another evidence to show that these methods are not compositional with respect to the underlying relations in an image, is our own experiences utilizing CLIP, the large language model we used in Figure 6 and L258-L265 of our paper. We show that the state-of-the-art vision-language model, i.e. CLIP, is not compositional with respect to the underlying relations in an image. We test the image-to-text retrieval performance of our proposed method and CLIP. We find that CLIP cannot understand spatial relations well, while EBMs can retrieve the correct descriptions.
>
> From a theoretical point of view, it also appears that using a language model to encode a relational scene description into a fixed vector cannot be compositional. This is due to the fact that while a compositional description can contain an arbitrary number of composed relations, the fixed dimensions latent vector only has a finite capacity and hence cannot represent all the compositional information.
>
>
>
> ---------------------
> **Q2: Related Work. As the authors mentioned in the related work section, the proposed idea has similarities with scene graphs. In the scene graph-to-image task, we model a set of triplets (<subject, relation, object>) as a structure and generate from it an image/video. I would like to mention [1], which model set of triplets (with the notion of time) to generate videos given a structure. Additionally, [2] generates videos given a set of descriptions the same as the authors proposed. I believe they could be mentioned.**
>
> We thank the reviewer for pointing out these 2 papers. These papers are indeed related to our work, as they also tackle the compositional generation. In the updated version of our paper, we will add these 2 papers to our discussion of related work.
>
> **Q3: Human eval. In my opinion, Tables 1 and 2 are missing human evaluation. Since generation and semantic equivalence are challenging to examine with current metrics, I suggest performing human analysis.**
>
> Please see our General Response Section 1.2 for the discussion of human evaluation.
>
> We do not believe that it makes sense to evaluate semantic equivalence using human analysis. This is because, in the semantic equivalence experiment, we utilize both real images and relational scene descriptions, and ask the model to predict which two relational scene descriptions are semantically closer given the input image. In particular, we construct two relational scene descriptions that match the image but describe the image in different ways, such as “a cabinet in front of a couch” and “a couch behind a cabinet”. There is one further description that does not match the image. Given an input image, the relative score difference between the two ground truth relational scene descriptions should be smaller than the difference between one ground truth relational scene description and the wrong relational scene description for the answer to be correct. Since the model is not generating any of the evaluated data, it does not make sense to perform the human evaluation.
>
>
> ---------------------------
> **Q4: Relation Descriptions Vs. Scene Graphs (SGs). I think it is interesting to understand the difference between relation descriptions-to-image and scene graphs-to-image tasks. Of course, both tasks aim to do the same, but it is still unclear which is a more appropriate direction for image generation, image editing, and generalization capabilities. Thus, I would expect to see a comparison of the proposed model and other existing SG-to-image methods.**
>
> Thanks a lot for your suggestion. We agree that comparing the relation descriptions and scene graphs is meaningful. In Table 1 of our supplemental material, we already show one baseline, i.e. StyleGAN2 (Scene Graph), that uses the graph network to encode objects and relations. The final representation of the scene graph is the concatenation of object embedding and relation embedding. Please see the supplemental material for more details.
> The image generation results on the Clevr dataset are listed in the table below. We compare the scene graph baseline, StyleGAN2 (Scene Graph), and our approaches on the 1R, 2R, and 3R test sets. We report the binary classification accuracy as we used in the paper. Our approaches outperform the baseline by a large margin.
>
> In addition, we added another baseline, i.e. sg2im [1], of using scene graphs in this rebuttal. Sg2im performs well on the 1R test set but fails on the more challenging test sets, such as 2R and 3R. Our approaches still outperform sg2im. These experiments indicate that using relation descriptions seems better than using scene graphs. In addition, relational scene descriptions are more flexible and easier to be composed than scene graphs.
>
> | Model				  |  1R Acc  |  2R Acc   |  3R Acc |
> | ----| ---- | ----- | ---- |
> |   StyleGAN2 (Scene Graph)       |    4.24     |    0.74      |    0.26 |
> |   Sg2im			  |    81.34   |    12.00    |    3.20 |
> |   Ours (CLIP)                              |    92.46   |     57.06    |   26.68 |
> |   Ours (Learned Embedding)      |    95.30   |    59.42    |   27.58 |
>
> *[1] Johnson, Justin, Agrim Gupta, and Li Fei-Fei. "Image generation from scene graphs." Proceedings of the IEEE conference on computer vision and pattern recognition. 2018.*
>
>
> **Human evaluation:**
>
> We further added a human evaluation to evaluate the image generation quality of sg2im and our approach (Ours (Learned Embedding)).
>
> We compare the correctness of the object relations in the generated images and the input language of our proposed model (Learned Embedding) and sg2im. Given a language description, we generate an image using our model (Learned Embedding) and sg2im. We shuffle these two generated images and ask the workers to tell which image has better quality and matches the object relations of the input language description. We tested 300 examples in total, including 100 examples with 1 sentence relational description (1R), 100 examples with 2 sentence relational descriptions (2R), and 100 examples with 3 sentence relational descriptions (3R). There are 32 workers involved in this human experiment.
>
> 1R results: The workers think there are 77% of examples that our proposed model (Learned Embedding) is better than sg2im.
>
> 2R results: The workers think there are 89% of examples that our proposed model (Learned Embedding) is better than sg2im.
>
> 3R results: The workers think there are 83% of examples that our proposed model (Learned Embedding) is better than sg2im.
>
> The human experiments further show that using relation descriptions is better than using scene graphs. The conclusion is coherent with our binary classification evaluation results.
> We will add these results in the updated version of our paper.

---

> ### Comment · Reviewer_TEjR · 2021-08-18
> **Final score**
>
> After reading the authors' feedback and other reviewers' opinions, I would like to thank the authors for their rebuttal.
>
> The rebuttal addresses my concerns. I believe this paper should be accepted since it maintains the high bar of the conference quality. I vote for 7.

---

### Official Review · Reviewer_vapy · 2021-07-16

**Rating:** 6
**Confidence:** 3

**Summary:**

This paper proposes to use EBM to model relations s in a factorized manner. The developed technique was applied to scene generation and experiments show that  a factorized decomposition allows more faithfully to both generate and edit scenes.

**Ethical Concerns:**

There's no obvious ethical issue.

**Limitations And Societal Impact:**

Yes they were adequately discussed.

I don't think the readers can learn much from the method section - it was simply re-introducing the same EBM as on textbooks. On the other hand, it is not clear how these components are connected. I would suggest the authors to refurbish the writing a bit.

**Main Review:**

- The novelty of the paper is potentially limited. Basically [1] used EBM to compose objects, while this paper uses EMB to compose relations. However, there's no fundamental difficulty in the adaption.
- Do the authors regard encoding the relation as a set (which I think is commonly used in knowledge graph) as the key innovation here? If not, I think it is necessary to add baselines such as (StyleGAN + set encoding). It is worth knowing how much improvement comes from changing encoding.
- Why is the baseline (StyleGAN + CLIP) better on iGibson compared with Clevr? It seems to me that iGibson has more complex scenes.
- The result looks pretty good, which accounts for my decision to suggest borderline accept.

[1] Du, Yilun, Shuang Li, and Igor Mordatch. "Compositional visual generation with energy based models." Advances in Neural Information Processing Systems 33 (2020): 6637-6647.

-------- post rebuttal update

I have checked the reviews and rebuttals of this submission. I lean towards acceptance. However, I still think the novelty of the submission is somewhat limited, so I keep my original rating as borderline accept.

**Time Spent Reviewing:**

2

---

> ### Author Response · Authors · 2021-08-10
> **Author Response to Reviewer vapy**
>
> **High-level response**
>
> Thank you for your detailed comments and feedback; we appreciate the time and attention you spent on reviewing our paper. We have addressed concerns about metrics below and have further attached additional real-world results. Please let us know if you have any additional questions. We are happy to clarify or provide additional experiments.
>
> ------------------
>
> **Q1: The novelty of the paper is potentially limited. Basically [1] used EBM to compose objects, while this paper uses EMB to compose relations. However, there's no fundamental difficulty in the adaption.**
>
> Please see our General Response Section 1.3.
>
> ------------------
>
> **Q2: Do the authors regard encoding the relation as a set (which I think is commonly used in knowledge graphs) as the key innovation here? If not, I think it is necessary to add baselines such as (StyleGAN + set encoding). It is worth knowing how much improvement comes from changing encoding.**
>
> The relations in our proposed method are not encoded as a set, but rather as a group of factorized EBMs. As suggested, we add a baseline, i.e. StyleGAN2 (set encoding), in this response. We encode the triplets (object 1, left, object 2) by simply adding embeddings of three items to have order-invariant property. Due to this property, StyleGAN2 (set encoding) performed poorly because it can’t differentiate (object 1, relation, object 2) from (object 2, relation, object1).
>
> We previously reported the performance of StyleGAN2 (Scene Graph) which also encodes a set of relations, albeit utilizing a graph neural network that iteratively encodes each relation (making individual relation embeddings sensitive to the order of objects).   The classification scores for StyleGAN2 (Scene Graph) were reported in the supplement of the original paper and are also worse than our model.
> We reported the same classification scores as we used in Table 1 in the main paper for comparison.
>
> The image generation results on the Clevr dataset are listed below:
>
> | Method                                   |    1R  Acc   |    2R Acc      |    3R Acc     |
> | ----| ----| ----| ----|
> | StyleGAN2 (Set Encoding)    |   4.12    |    0.70     |   0.14  |
> | StyleGAN2 (Scene Graph)     |   4.24    |    0.74     |   0.26  |
> | Ours (Learned Emb)               |  95.30  |   59.42   |  27.58   |
>
> Our method is much better than the new baseline.
>
> ------------------
>
> **Q3: Why is the baseline (StyleGAN + CLIP) better on iGibson compared with Clevr? It seems to me that iGibson has more complex scenes.**
>
> The iGibson scenes might look more complex since their background is harder than the Clevr scenes. However, the number of object attributes, the types of object relations, and the object combinations in the iGibson scenes are fewer than the Clevr scenes. For example, each scene on the iGibson dataset contains 1 ∼ 3 objects while each Clevr scene contains 1 ∼ 5 objects. Please see L166-L173 in the main paper for the descriptions of the iGibson and Clevr datasets.
>
> In terms of the baseline (StyleGAN + CLIP) is better on iGibson compared with Clevr. This is because the object attributes and relations on the iGibson dataset are simpler than the Clevr dataset as described above. StyleGAN2 (CLIP) can perform well on the easier test set, such as the 1R test set for the iGibson dataset. The 1R test set is simple as it generates images based on a single relational description and it has the same data distribution as the training set. However, when it comes to more challenging test sets, such as 2R, 3R, or Clevr, our proposed model has better compositionality and generalization ability and it performs much better than the baselines.
>
> ------------------
>
> **Q4: I don't think the readers can learn much from the method section - it was simply re-introducing the same EBM as on textbooks. On the other hand, it is not clear how these components are connected. I would suggest the authors to refurbish the writing a bit.**
>
> Thanks a lot for your suggestion. We will update the method section and add more details on both the specific construction of relational EBMs and how they are directly applied to compose visual relations.

---

### Official Review · Reviewer_JivJ · 2021-07-16

**Rating:** 6
**Confidence:** 2

**Summary:**

This paper proposed image generation and image editing methods using the energy-based model for learning relational interaction. The proposed method can be summarized as : starting from N discrete relation and their features can be represented as a probability distribution function. This probability distribution is modeled using an energy-based model.  An energy-based model is an unnormalized probability distribution and trained with contrastive divergence. Monte Carlo simulation has been used iteratively to sample from the probability distribution. Afterward, a text encoder has been used to model relation triplet for every relation.

Authors have conducted experiments on two datasets Clever and iGibsion for multiple task categories.  In addition, a brief ablation experiment has been done for semantic equivalence and zero-shot generalization.

**Limitations And Societal Impact:**

There are a few limitations of the paper :
1. Experiment conducted on the synthetic dataset, this raises the doubt its performance on real relational dataset like visual genome.
2. How the interdependency of the relationship can be exploited using this framework or can be extended?

**Main Review:**

The paper is clearly written and easy to understand. The proposed energy-based model on image generation is relatively novel, but its application is somewhat limited due to the lack of global relation context.

**Time Spent Reviewing:**

4

---

> ### Author Response · Authors · 2021-08-10
> **Author Response to Reviewer JivJ**
>
> **High-level response**
>
> Dear reviewer, thank you very much for your detailed comments and feedback. We have added additional real-world results and have provided textual clarifications. We will incorporate all of them in the final version. Please feel free to let us know if you have more questions about the paper. We will try our best to address your concerns.
>
> -------------------
> **Q1: Experiment conducted on the synthetic dataset, this raises the doubt of its performance on real relational datasets like visual genome.**
>
> Please see our General Response Section 1.1.
>
> -------------------
> **Q2: How the interdependency of the relationship can be exploited using this framework or can be extended?**
>
> We note that our approach towards composing EBMs allows us to model the interdependence between separate relations. In particular, consider the relation $r1$ which specifies that “a blue cube is above a red cube”, and relation $r2$ which specifies “a red cube is to the right of a green cube”. In the absence of relation $r2$, $E(x|r1)$ will assign low energies to all images that exhibit relation $r1$, while in the absence of relation $r1$, $E(x|r2)$ assign low energies to all images that exhibit relation $r2$. If we sum up the two energy functions $E(x|r1) + E(x|r2)$, this function will assign low energies to only images that exhibit both relation $r1$ and relation $r2$, as all other images will have high energies under either $E(x|r1)$ or $E(x|r2)$. Thus, directly summing up the energy of relations in an image captures the interdependence between these relations, and ensures that generated images satisfy all relations.

---

> ### Comment · Reviewer_JivJ · 2021-08-12
> **Rebuttal Review**
>
> Limitations and doubts raised during the review are clearly addressed by the authors.

---

### Official Review · Reviewer_ubTt · 2021-07-17

**Rating:** 7
**Confidence:** 3

**Summary:**

The paper introduces an energy-based model to compose separate relations in a factorized manner.  A relational scene description is decomposed into individual relations, and modeled as the product of individual probability distributions across relations. The proposed method is shown to be able to generate images with good quality from the relational descriptions and understand the semantic differences between relational descriptions. Zero-shot evaluation also suggest that the proposed approach can generalize to unseen relational descriptions.

**Limitations And Societal Impact:**

Yes.

**Main Review:**

Overall, I think this paper conduct a comprehensive evaluation of the proposed method and demonstrate its power on several tasks: image generation, image editing, relational understanding. The model performance is strong, the proposed method outperforms StyleGAN2 for almost all cases both quantitatively and qualitatively.

I would like to learn from the authors about the following questions.

`Quantitative Comparison for Image Generation`

Is it a standard practice to perform binary classification on the generated images? How should we interpret the accuracy? Could a model somewhat trick this score? For example, in the qualitative analysis, "ours" seem to do better than the baselines, but the accuracy is lower.

In [1], the author proposed to use object detection metrics as the quantitative measure. And a graph-based relational similarity score is used to test the correct placement of objects, without requiring the model to draw the objects in the same exact locations. Would these more finegrained metric be better than a global match through a binary classifier between image and relational scene descriptions?

Also, human evaluation on the generated images to compare different model performance might be a more convincing quantitative measure.

[1] Keep Drawing It: Iterative language-based image generation and editing: https://nips2018vigil.github.io/static/papers/accepted/13.pdf

`Application to more realistic scenes`

Could the method be extended to more realistic scenes other than the artificial ones in CLVER and iGibson?

I would imagine there are several challenges here. One is the lack of such detailed relation descriptions for real images, especially for complex scenes. In addition, the common practice to quantitatively evaluate image generation results for real scenes is FID or inception score.  Whether the proposed method can improve FID or inception score is unclear. Third, there will be more complex relation/attribute/object for real scenes, larger vocabulary size for the learned embedding. My guess is CLIP text encoder might perform better, as it has seen large scale image-text pairs that might cover a wide range of relation/attribute/object. I would be happy to hear authors' thoughts on this.



**Time Spent Reviewing:**

2

---

> ### Author Response · Authors · 2021-08-10
> **Author Response to Reviewer ubTt**
>
> **High-level response**
>
> Thank you for your detailed comments and feedback; we appreciate the time and attention you spent on reviewing our paper. We have addressed concerns about metrics below and have further attached additional real-world results. Please let us know if you have any additional questions. We are happy to clarify or provide additional experiments.
>
> ------------------
>
> **Q1:  Is it a standard practice to perform binary classification on the generated images? How should we interpret the accuracy?**
>
> Thanks a lot for sharing your concerns and suggestions. The main reason we choose this binary classification is that 1) it has been used in previous works for compositional image generation Du et al. [1] and [2]; 2) it is hard to find a better automatic evaluation for this task as there are no ground truth relations in the generated images to be compared with; 3) the binary classification is reliable most of the time.
>
> We agree that the binary classification might not be a perfect evaluation, as the pre-trained classifier might generate false predictions. However, we think the binary classification is still reliable enough to evaluate the performance of different methods.
> The binary classification is trained to predict whether the given input language description and image are matching or not. It first outputs a matching score of the images containing the context that is described in the language. This matching score is then used to do binary classification. We ensure that the classifier is well-trained so that it can correctly predict whether an image and a language are matching or not (the accuracy on real images is close to 100%).
>
> During testing, we feed in the language and the corresponding generated image into the binary classifier. The higher the accuracy is, the image is more likely to contain the context described in the language, which means the generated images are good (match the language).
>
> We further add results of using human evaluation in our General Response Section 1.2. The conclusion obtained from the human evaluation is coherent with our binary classification evaluation results.
>
>
> *[1] Du, Yilun, Shuang Li, and Igor Mordatch. "Compositional visual generation with energy-based models." Advances in Neural Information Processing Systems 33 (2020): 6637-6647.*
>
> *[2] Du, Yilun, Shuang Li, Joshua Tenenbaum, and Igor Mordatch. "Improved Contrastive Divergence Training of Energy-Based Models Supplementary." ICML (2021)*
>
> ------------------
>
> **Q2:  Could a model somewhat trick this score? For example, in the qualitative analysis, "ours" seem to do better than the baselines, but the accuracy is lower.**
>
> As we have explained in the last response, we agree that the binary classification might not be a perfect evaluation, as the pre-trained classifier might generate false predictions. However, we think the binary classification is still reliable enough to evaluate the performance of different methods as we have explained in the last response. The conclusion obtained from the human evaluation is also coherent with our binary classification evaluation results.
>
>
> In terms of the binary classification accuracy on image generation, our approach (Learned Embedding) seems worse than StyleGAN2 (CLIP) on the iGibson dataset (1R). This is because the 1R test set is simple and the data distribution of the 1R test set is the same as the training set. On the other hand, the object attributes and relations on the iGibson dataset are simpler than the Clevr dataset. The number of object attributes, the types of object relations, and the object combinations in the iGibson scenes are fewer than the Clevr scenes. For example, each scene on the iGibson dataset contains 1 ∼ 3 objects while each Clevr scene contains 1 ∼ 5 objects. Please see L166-L173 in the main paper for the descriptions of the iGibson and Clevr datasets.
> StyleGAN2 (CLIP) can perform well on the simpler 1R test set on the iGibson dataset. However, when it comes to more challenging test sets, such as 2R, 3R, or Clevr, our proposed model has better compositionality and generalization ability and performs much better than the baselines.
>
> For the qualitative analysis of image generation in Figure 4 on the iGibson dataset, both the 1R results (the left part of Figure 4) of our approach (Learned Embedding) and StyleGAN2 (CLIP) are correct. But for the 2R results  (the right part of Figure 4), our approach is much better than StyleGAN2 (CLIP). This result is coherent with the binary classification results.
>
> ------------------
>
> **Q3: In [1], the author proposed to use object detection metrics as the quantitative measure. And a graph-based relational similarity score is used to test the correct placement of objects, without requiring the model to draw the objects in the same exact locations. Would these more fine-grained metrics be better than a global match through a binary classifier between image and relational scene descriptions?**
>
> We agree that the evaluation metric used in [1] has its advantages. For example, it can construct scene graphs for both generated and ground truth images without telling the model to precisely draw objects at the exact locations. However, the evaluation metric used in [1] has its disadvantages as well. For example, it heavily relies on the pre-trained object detector and localizer. The pre-trained object detector or localizer could generate false predictions on both real images and generated images, especially when the generated images are out of the training distribution.
>
> In this rebuttal, we reported two baselines and our approach using the evaluation metric [1] suggested by the reviewer. The image generation results on the Clevr dataset of our approach and baselines are listed in the table below. The conclusion obtained by using this new metric is coherent with using our binary classification metric that our proposed method outperforms the baselines.
>
> The evaluation metric used in [1] focuses more on local matching, while ours focuses on global matching. We think it is good to report both metrics in the updated version of our paper. We will also incorporate the human evaluation in the updated version.
>
> | Model | 1R Acc | 2R Acc | 3R Acc|
> | -----| ------| ------- | ----- |
> |StyleGAN2 | 22.37 | 19.75 | 17.13|
> |StyleGAN2 (CLIP) | 37.50 | 28.62 | 28.75|
> |Ours (Learned Embedding) | 50.77 | 36.87 | 42.50
>
> ------------------
>
> **Q4: Human evaluation on the generated images to compare different model performance might be a more convincing quantitative measure.**
>
> Please see our General Response Section 1.2.
>
> ------------------
>
> **Q5: Could the method be extended to more realistic scenes other than the artificial ones in CLVER and iGibson?**
>
> Please see our General Response Section 1.1.

---

> ### Comment · Reviewer_ubTt · 2021-08-31
> **Rebuttal Review**
>
> Thanks the authors for addressing my concerns and comments. After reading other reviewers' comments and full rebuttal, I would like keep my rating as 7.
>
> I especially like the discussion about different quantitative measures and the human evaluation on generated images, I suggest to add them into the final version.

---

### Official Review · Reviewer_7kfr · 2021-07-19

**Rating:** 8
**Confidence:** 4

**Summary:**

This work proposes to represent individual visual relations in an image as unnormalized probabilistic densities following the scheme of Energy-Based Models, by which the modeling of each relation can be factorized and enables their composition to generate images. According to the reported experiments, this kind of relation description can help generate and edit images with multiple composed relations and understand semantically equivalent relational scene descriptions, which is also generalizable to unseen dataset during training.

**Limitations And Societal Impact:**

Yes, the authors have discussed the limitations and potential negative societal impact of this work in the section of conclusion.

**Main Review:**

This paper employs the EBM to model composed relations, as a product of individual learned probability distributions for each relation description, thus offers more flexibility in generating images with multiple composed visual relations. Though the experiments were conducted upon synthetic and somewhat simple datasets, this approach shows certain potentials in understanding semantically equivalent relations, which is quite hard by previous methods and can be generalized to unseen scenarios.

Overall, this study is helpful to researchers in the fields of image generation and visual understanding. This paper is well written, clearly illustrated, and appropriately structured.

In my opinion, the proposed method inherits the main merits from recent EBM-based image generation methods, that try to employ EBM to learn how to compose entities or other elements in the scene (Du et al. 2019, Du et al. 2020, etc.)  Thus this paper may be less significant if it looks like an extension to these prior arts (even the model architectures are similar to those from Du et al. 2021). I acknowledge that this paper has validates that the EBM models can compose visual relations in image generation, but a comprehensive discussion about the technical uniqueness of the proposed method would be encouraging.

I have a little bit of concern about the effectiveness of the proposed method since the testing scenarios were simple and a little bit far away from generating real scenes.

Moreover, since the language parser converts a textual description into a series of relation descriptions, I am curious whether the language parser can disambiguate multiple entities from the same categories and attributes? And then whether the proposed method can generate multiple similar objects rather than one object, if the relation descriptions do not mark the identities of these objects.

The references require significant modification, about the format, missing items, duplicated entries, etc.

If possible, adding algorithm snippets would help the readers to understand the training/testing pipeline.

**Time Spent Reviewing:**

3 hours

---

> ### Author Response · Authors · 2021-08-10
> **Author Response to Reviewer 7kfr**
>
> **High-level response**
>
> Thank you for your detailed comments and feedback; we appreciate the time and attention you gave to reviewing our paper. We have addressed concerns about novelty, real image results, and textual clarifications. Please let us know if you have any additional questions. We are happy to clarify or provide additional experiments.
>
> ---------------
> **Q1:  I acknowledge that this paper has validates that the EBM models can compose visual relations in image generation, but a comprehensive discussion about the technical uniqueness of the proposed method would be encouraging.**
>
> Please see our General Response Section 1.3.
>
>
> ---------------
> **Q2: I have a little bit of concern about the effectiveness of the proposed method since the testing scenarios were simple and a little bit far away from generating real scenes.**
>
> Please see our General Response Section 1.2.
>
>
> ---------------
> **Q3: Whether the language parser can disambiguate multiple entities from the same categories and attributes? And then whether the proposed method can generate multiple similar objects rather than one object, if the relation descriptions do not mark the identities of these objects**
>
> Yes, the language parser can disambiguate multiple entities from the same categories and attributes.
>
> Yes, our method can generate multiple similar objects even if the relation descriptions do not mark the identities of these objects.
>
> We use the Stanford CoreNLP parser to extract the part-of-speech tags and dependencies of scene relation descriptions [1]. We then utilize the extracted part-of-speech tags and dependencies to obtain the object attributes and their relations. For example, when we send “a small blue metal cube to the left of a small blue metal cube” to the language parser, we get a parsing tree that contains the tags and dependencies of the words, such as nouns (“cube”), adjectival modifiers (“small” → “cube”), nominal modifiers (“cube” → “left”), etc. We use this information to get the object attributes and their relations and use them as the input of our model.
>
> In terms of image generation results, we test our model to generate multiple similar objects when the relation descriptions do not mark the identities of these objects. The language descriptions and corresponding generated images can be found using this link:
> https://composevisualrelations.github.io/#similar_object_generation
>
> Our model can generate multiple similar objects rather than one object, even if the relation descriptions do not mark the identities of these objects.
>
> [1] https://stanfordnlp.github.io/CoreNLP/
>
>
> ---------------
> **Q4: The references require significant modification, about the format, missing items, duplicated entries, etc.**
>
> Thanks a lot for your suggestion. We will correct the format, add missing items, and remove the duplicated entries in the final version.
>
> ---------------
> **Q5: If possible, adding algorithm snippets would help the readers to understand the training/testing pipeline.**
>
> Thanks a lot for the suggestion. We have added the training and testing algorithm snippets in this rebuttal. Please find then using this link: https://composevisualrelations.github.io/#algorithms
>
> We will incorporate these algorithm snippets in the updated version of the paper.

---

> ### Comment · Reviewer_7kfr · 2021-08-31
> **Post-rebuttal Comments**
>
> I have checked the reviews and rebuttals of this submission. Most concerns, such as technical uniqueness, quantitative evaluations and subjective testing, have been addressed. Though this method may not reliably generate images in real scenarios, but it indeed indicates some sorts of composability. So I keep my rating as 8.

---

### Author Response · Authors · 2021-08-10
**General Response**

We thank the thorough and insightful comments from all the reviewers. The reviewers noted that our framework is novel, the performance of our proposed model is strong, and our study is helpful to researchers in the fields of image generation and visual understanding. Reviewer 7kfr further pointed out that our approach can understand semantically equivalent relations, which are quite hard by previous methods. Reviewer 7kfr and Reviewer TEjR further thought it is compelling that our approach can be generalized to unseen scenarios. Reviewer ubTt noted that our evaluation is also comprehensive.

Each individual reviewer further has their own personalized questions about the underlying text and statements of the paper. We have answered the common questions below and also provide detailed discussions of evaluation results in the individual reviewer comments, as well as additional experiments.

We showcase our additional qualitative results at the webpage URL https://composevisualrelations.github.io/.

--------------------------------

## 1.1 Real Scenes/datasets (Reviewer 7kfr, ubTt, JivJ)

In terms of image generation results on real scenes, we train and evaluate our model on two real-world datasets, one real-world block dataset and the visual genome dataset.

The real-world block dataset is from [1] and we train our model using the object relations, e.g. “above” and “below”. The language descriptions and corresponding generated images can be found using this link:  (https://composevisualrelations.github.io/#block_dataset). We show the images generated conditioned on 1 relational description, 2 relational descriptions, and 3 relational descriptions, respectively.

For the visual genome dataset, we train our models on a subset that consists of common objects and relations for computational efficiency. The language descriptions and corresponding generated images can be found using this link: (https://composevisualrelations.github.io/#vg_image_generation). We agree with **Reviewer ubTt** that the CLIP text encoder performs better, as it has seen large-scale image-text pairs that cover a wide range of relations/attributes/objects.

Our approach is able to generate images (objects and their relations) matching the given language descriptions on the real-world block dataset and the visual genome dataset. The quality of generated images on the real-world block dataset is great. However, the quality of the visual genome is a bit worse. We believe, given more time and more computation resources, we can further improve the generation quality.

We would also like to clarify that the main focus of this paper is “Learning to Compose Visual Relations”. Our model has shown good compositionality ability and generalization ability on several datasets. As we have mentioned in our limitation section, a good direction for future work would be to study how these models scale to complex datasets found in the real world.


*[1] Lerer, Adam, Sam Gross, and Rob Fergus. "Learning physical intuition of block towers by example." International conference on machine learning. PMLR, 2016.*

--------------------------------

## 1.2 Human evaluation (Reviewer ubTt, TEjR)

We agree that human evaluation is more accurate than using binary classification. Thus in this rebuttal, we have added a human evaluation to evaluate the image generation quality.

We compare the correctness of the object relations in the generated images and the input language of our proposed model (Learned Embedding) and StyleGAN2 (CLIP). Given a language description, we generate an image using our model (Learned Embedding) and StyleGAN2 (CLIP). We shuffle these two generated images and ask the workers to tell which image has better quality and the object relations match the input language description. We tested 300 examples in total, including 100 examples with 1 sentence relational description (1R), 100 examples with 2 sentence relational descriptions (2R), and 100 examples with 3 sentence relational descriptions (3R). There are 32 workers involved in this human experiment.

1R results: The workers think there are 87% of examples that our proposed model (Learned Embedding) is better than StyleGAN2 (CLIP).

2R results: The workers think there are 86% of examples that our proposed model (Learned Embedding) is better than StyleGAN2 (CLIP).

3R results: The workers think there are 91% of examples that our proposed model (Learned Embedding) is better than StyleGAN2 (CLIP).

The human experiment shows that our proposed method is better than StyleGAN2 (CLIP) (the best baseline approach). The conclusion is coherent with our binary classification evaluation results.

--------------------------------

## 1.3 Novelty (Reviewer 7kfr, vapy)

Our primary contribution in this paper is in the combination of vision and language for compositional image generation/editing. We are interested in constructing an underlying system that can robustly understand a relational scene description, and generate or edit images to exhibit the desired scene descriptions. We believe that such a task is important to solve, and occurs frequently in real-world settings, such as robotics.

While there are a variety of approaches have been proposed towards this task, ranging from the holistic description, such as CLIP, to a scene graph encoding, our major contribution in this paper is showing that by representing individual relations as unnormalized probability densities, and by composing them together, we are able to construct a system that can robustly understand relational scene descriptions.

On the technical side, while there were no significant changes to the underlying EBMs training algorithm, utilizing EBMs to represent individual relations in the vision-language domain poses several modeling challenges compared to past works. First, while prior works towards composing EBMs typically assume a fixed number of factors that describe an underlying object, with individual EBMs learned for each factor. There are a large number of possible visual relations, making the previous approach intractable. Furthermore, while past composed factors are now specified with respect to an entire scene, in the visual relation setting, each composed factor is specified with respect to a set of underlying objects, leading to an additional complication on the implementation of such object conditioning.  To solve these issues, we utilize the approach described in the supplementary material Section 4, where an embedding is utilized to encode a specific relation, and a separate embedding scheme is utilized to encode the relation to particular objects.

We will clarify the importance of such changes and further highlight its difference from previous works in the updated version of the paper.

--------------------------------

## Conclusion

We further address the questions proposed by each individual reviewer below. Please feel free to let us know if you have more questions about the paper. We will try our best to address your concerns.

---

### Decision · Program_Chairs · 2021-09-27

**Decision:**

Accept (Spotlight)

**Comment:**

This paper presents a quite novel framework to factorize and compose separate object relations, with the ability to generate and edit images with composed relations. The performance is significantly better than StyleGAN2 CLIP and encouraging. One of the limitations that reviewers are concerned about is that most of the experiments are conducted on synthetic data, and more analyses on real-world scenes are expected. All reviewers carefully read the authors' responses, have extensive discussions and agree on the novelty and explanation of this work. Considering the high quality of this paper and consistent positive comments by reviewers, The AC recommends accepting this paper.